# Freelance Holography, Part I:

## Setting Boundary Conditions Free in Gauge/Gravity Correspondence

**A. Parvizi**[a] , **M.M. Sheikh-Jabbari**[b] , **V. Taghiloo**[b,c]

[a] *University of Wroclaw, Faculty of Physics and Astronomy, Institute of Theoretical Physics,*
*Maksa Borna 9, PL-50-204 Wroclaw, Poland*

[b] *School of Physics, Institute for Research in Fundamental Sciences (IPM),*
*P.O.Box 19395-5531, Tehran, Iran*

[c] *Department of Physics, Institute for Advanced Studies in Basic Sciences (IASBS),*
*P.O. Box 45137-66731, Zanjan, Iran*

*E-mail:* aliasghar.parvizi@uwr.edu.pl, jabbari@theory.ipm.ac.ir,
v.taghiloo@iasbs.ac.ir

ABSTRACT: We explore AdS/CFT duality in the large $N$ limit, where the duality reduces to gauge/gravity correspondence, from the viewpoint of covariant phase space formalism (CPSF). In particular, we elucidate the role of the $W, Y$, and $Z$ freedoms (also known as ambiguities) in the CPSF and their meaning in the gauge/gravity correspondence. We show that $W$-freedom is associated with the choice of boundary conditions and slicing of solution space in the gravity side, which has been related to deformations by multi-trace operators in the gauge theory side. The gauge/gravity correspondence implies the equivalence of on-shell symplectic potentials on both sides, thereby the $Y$-freedom of the gravity side specifies the on-shell symplectic form of the gauge theory side. The $Z$-freedom, which determines the corner Lagrangian on the gravity side, establishes the boundary conditions and choice of slicing in the boundary theory and its solution space. We utilize these results to systematically formulate freelance holography in which boundary conditions of the fields on the gravity side are chosen freely and are not limited to Dirichlet boundary conditions and discuss some examples with different boundary conditions.

## 1   Introduction

The Covariant Phase Space Formalism (CPSF) [1–4] (for review see [5–7]) provides a rigorous framework for associating charges with the symmetries of a theory, starting from its Lagrangian. This formalism involves two key manifolds: the *spacetime* manifold over which the theory is defined and the *solution phase space*. The spacetime is typically a manifold with boundary. The solution phase space, as the name suggests, is equipped with a symplectic two-form, which serves as a fundamental quantity for computing charges associated with symmetries, including the global or boundary part of gauge symmetries and

diffeomorphisms. However, the formalism introduces certain freedoms (or ambiguities), denoted by $Y$, $W$, and $Z$, which influence the definitions of the symplectic potential, symplectic form, charges, charge algebra, and other related quantities [3, 6]. To obtain unique and unambiguous results and definitions for these quantities and structures, it is necessary to fix these freedoms. Achieving this requires additional physical criteria or constraints.

The AdS/CFT duality [8, 9], also known as holographic duality, provides a formulation of quantum gravity in an asymptotically AdS space (the bulk side) in terms of a conformal field theory (CFT) that resides in one lower spacetime dimension, typically on the causal (conformal) boundary of the AdS space (the boundary side). When dealing with a theory on a manifold with a boundary, CPSF offers a natural framework for studying the duality. This leads us to our central question: *What is the significance of these freedoms in the context of holography? Moreover, does holography provide a way to fix these freedoms?* In this paper, we explore these questions and demonstrate how the AdS/CFT duality resolves these freedoms within the CPSF.

Let us examine these freedoms individually, beginning with $W$-freedom. This freedom corresponds to the choice of boundary terms (total derivative terms) in the Lagrangians. It is well known that the variational principle relates these boundary terms to the boundary condition of the fields. A seminal example is the Gibbons-Hawking-York term [10, 11], which demonstrates that adding a boundary Lagrangian to the Einstein-Hilbert action is essential for a well-defined action principle with Dirichlet boundary conditions in general relativity. Another example appears in the context of holographic renormalization [12–17], where appropriate counterterms must be added to the action to ensure finite physical quantities at infinity, such as the on-shell action. For a procedure to regularize surface charges, see [18]. Similarly, in holography, defining a suitable energy-momentum tensor for the boundary theory requires adding counterterms to normalize the energy-momentum tensor [19]. Additionally, in the Hamiltonian and Hamilton-Jacobi formalisms, physical quantities are typically renormalized using techniques such as vacuum subtraction [20]. All of these methods and adjustments can be understood as manifestations of $W$-freedom. In this work, we explore these established interpretations and applications of $W$-freedom, consolidate these insights, and introduce new perspectives on its meaning and utility.

It is important to note that even after renormalizing our physical quantities, a residual freedom remains, allowing for the addition of an extra $W$-term. While this preserves the finiteness of physical quantities, it modifies their finite values. This raises a key question: what is the physical significance of these extra components of $W$-freedom? We demonstrate that this aspect of $W$-freedom is determined by the boundary conditions of the fields in the bulk theory and serves as the generator of the *change of slicing* (or canonical transformations) in the solution phase space of the bulk theory [21–26]. Within the AdS/CFT framework, as mentioned above, $W$-freedom arises in holographic renormalization [12] as counterterms in the bulk theory while simultaneously renormalizing physical observables in the boundary theory. Furthermore, we demonstrate that $W$-freedom, which modifies the bulk boundary conditions, corresponds to a multi-trace deformation in the boundary theory [27–29] (and for a systematic treatment of multi-trace deformations see also [30–33] and also [34–51]).

In field theories defined on spacetimes with boundaries, there exist not only bulk degrees of freedom (dof) that propagate throughout the bulk but also boundary degrees of freedom that reside on the spacetime boundary. For discussions on boundary modes, see [22, 52–62]; for edge modes, see [63–69]; and for examples from condensed matter physics, see [70–73]. The CPSF provides the symplectic form for the bulk modes, while the symplectic form of the boundary degrees of freedom is determined by fixing the $Y$-freedom. We establish that, as expected, the $Y$-freedom of the bulk theory corresponds to the

symplectic potential of the dual boundary theory. The $W$ and $Y$ freedoms overlap in what we refer to as $Z$-freedom, making it not entirely independent of the other two. While $W$-freedom determines the boundary Lagrangian, $Z$-freedom governs the corner Lagrangian. Consequently, the $Z$-freedom of the bulk theory dictates both the boundary conditions and the choice of slicing in the solution phase space of the dual boundary theory.

The covariant phase space view of the gauge/gravity correspondence paves the way to extend the correspondence by providing a systematic way to relax the bulk field boundary conditions from the usual Dirichlet boundary conditions to more general cases, hence formulating a *freelance holography* in which boundary conditions of the bulk fields are relaxed. In this sense our work provides extends the analysis in [33] to more general boundary conditions and to beyond three dimensional bulk theories.

**Outline of the paper.** In Section 2, we review the core principles of the Covariant Phase Space Formalism (CPSF) and the gauge/gravity correspondence. In Sections 3 and 4, we analyze the gauge/gravity correspondence through the lens of CPSF and investigate the role of CPSF freedoms on each side of the holographic duality. Sections 5 and 6 contain applications of these results to the formulation of freelance holography. We systematically extend the conventional AdS/CFT framework, which imposes Dirichlet boundary conditions on gravity fields, to include Neumann, conformal, and a new class of boundary conditions where the boundary metric is specified up to a conformal factor that depends on the trace of the Brown-York energy-momentum tensor. Section 7 is devoted to a summary of the results and an outlook. In Appendix A, we review the construction of solution space for the free scalar theory in AdS space.

## 2 Review of basic formalisms

To set the stage and fix conventions, we first review the two basic formulations we are going to correlate: Covariant Phase Space Formalism (CPSF) and the gauge/gravity correspondence.

### 2.1 Quick review of covariant phase space formalism

To set the conventions and make the paper more self-contained, we very briefly review the basics of the covariant phase space formalism that is well presented in the notion of $(p, q)$-forms, $p$-forms of the spacetime and $q$-forms over the solution space. We use boldface symbols to denote the $(p, q)$-forms. Lagrangian of any $D$ dimensional theory $\mathbf{L}$, is a top form, i.e., a $(D, 0)$-from. Therefore,

$$\mathrm{d}\mathbf{L} = 0, \qquad \delta\mathbf{L} \stackrel{\circ}{=} \mathrm{d}\mathbf{\Theta}, \tag{2.1}$$

where $\stackrel{\circ}{=}$ denotes on-shell equality and d and $\delta$ are exterior derivatives over the spacetime and solution space respectively. $\mathbf{\Theta}$ is the symplectic potential which is a $(D-1, 1)$-form. From the above and using Poincaré lemma over the solution space one learns that,

$$\mathrm{d}\delta\mathbf{\Theta} \stackrel{\circ}{=} 0 \quad \implies \quad \mathbf{\Theta} = \mathbf{\Theta}_{\mathrm{D}} + \delta\mathbf{W} + \mathrm{d}\mathbf{Y} + \mathrm{d}\delta\mathbf{Z}, \tag{2.2}$$

where $\mathbf{\Theta}_{\mathrm{D}}$ is a particular solution to $\mathrm{d}\delta\mathbf{\Theta}_{\mathrm{D}} = \delta\mathrm{d}\mathbf{\Theta}_{\mathrm{D}} = 0$, chosen to enforce the Dirichlet boundary condition for the field of the theory while ensuring the finiteness of the on-shell action and canonical variables (see Sections 5 and A for concrete examples). $\mathbf{W}, \mathbf{Y}, \mathbf{Z}$ are respectively $(D-1, 0)$, $(D-2, 1)$ and $(D-2, 0)$-forms, are integration constants, the three freedoms that will be specified by other physical requirements [3, 6]. One can in fact define the $(D-1, 1)$-form symplectic potential by $\mathrm{d}\delta\mathbf{\Theta} \stackrel{\circ}{=} 0$ without starting from (2.1). We note that $\mathbf{Z}$ may be absorbed in $\mathbf{Y}$ or $\mathbf{W}$ by calling $\mathbf{Y} + \delta\mathbf{Z}$ as the new $\mathbf{Y}$ or

taking $\mathbf{W} + \mathrm{d}\mathbf{Z}$ as the new $\mathbf{W}$. In other words, there is a freedom in defining $\mathbf{Y}, \mathbf{W}$, and there is an overlap between $\mathbf{Y}, \mathbf{W}$ terms, which we have explicitly extracted out in $\mathbf{Z}$: $\mathbf{Z}$ is the $\delta$-exact part of $\mathbf{Y}$ or the d-exact part of $\mathbf{W}$.

The symplectic density $\boldsymbol{\omega}$ which is a $(D - 1, 2)$-form, is defined as

$$\boldsymbol{\omega} := \delta\boldsymbol{\Theta} = \delta\boldsymbol{\Theta}_{\mathrm{D}} + \mathrm{d}\delta\mathbf{Y}, \qquad \mathrm{d}\boldsymbol{\omega} \overset{\circ}{=} 0. \tag{2.3}$$

Integrating $\boldsymbol{\omega}$ over a codimension 1 surface $\Sigma$ we obtain the symplectic form $\boldsymbol{\Omega}$

$$\boldsymbol{\Omega} := \int_\Sigma \boldsymbol{\omega} = \int_\Sigma \delta\boldsymbol{\Theta}_{\mathrm{D}} + \int_{\partial\Sigma} \delta\mathbf{Y}. \tag{2.4}$$

The symplectic form $\boldsymbol{\Omega}$ is a closed $(2, 0)$-form, $\delta\boldsymbol{\Omega} = 0$. In addition, the above shows that $\boldsymbol{\Omega}$ consists of codimension 1 and codimension 2 parts. Theories residing on spacetimes with boundaries generically have boundary modes as well as bulk modes, the bulk modes appear in the codimension 1 part of the symplectic form and the boundary modes in the codimension 2 part [60, 74]. Note that $\mathbf{W}, \mathbf{Z}$ freedom in the symplectic potential $\boldsymbol{\Theta}$ does not appear in the symplectic form and that the $\mathbf{Y}$-freedom contributes to the codimension 2 part of the symplectic form.

**Physical meaning of $\mathbf{W}, \mathbf{Y}, \mathbf{Z}$ freedoms in gravity.** The $\mathbf{W}$-freedom may be related to the change of Lagrangian by a total derivative. That is, upon the shift $\mathbf{L} \to \mathbf{L} + \mathrm{d}\mathbf{W}$, (2.1) implies $\boldsymbol{\Theta} \to \boldsymbol{\Theta} + \delta\mathbf{W}$. Explicitly, $\mathbf{W}$ is set by, or sets the, boundary Lagrangian. In a similar manner the $(D - 2, 0)$-form $\mathbf{Z}$-freedom contributes to "corner" Lagrangian, Lagrangian on the boundary of the boundary. Boundary and corner Lagrangians do not change the (bulk) equations of motion and the symplectic potential; their role is to set the boundary conditions on the bulk fields and to generate change of slicings on the solution phase space. The prime example of such a boundary term is the Gibbons-Hawking-York term which sets Dirichlet boundary conditions for metric components [10, 11]. This would be an essential point in discussions of the next section regarding gauge/gravity correspondence.

The $\mathbf{Y}$-freedom, however, has no direct appearance in the bulk theory and its Lagrangian $\mathbf{L} + \mathrm{d}\mathbf{W}$. $\mathbf{Y}$ plays the role of symplectic potential for the theory governing boundary degrees of freedom which resides on the codimension 1 boundary. This boundary symplectic potential and hence symplectic form for the boundary modes is specified once we fix $\mathbf{Y}$. Similarly, $\mathbf{W}$ freedom specifies the Lagrangian for boundary modes. In other words, $\mathbf{Y}, \mathbf{W}$ respectively specify the symplectic potential and Lagrangian for the boundary modes and $\mathbf{Z}$ generates change of slicings on the boundary sector of the solution phase space.

We should emphasize that in the discussions above the bulk and boundary just refers to what we have for any field theory residing on a $D$ dimensional spacetime with a $D - 1$ dimensional boundary; bulk and boundary in the above does not correspond to holographic pairs. In the holographic context either of the two sides have their own $\mathbf{W}, \mathbf{Y}, \mathbf{Z}$ which will be identified with corresponding indices, as discussed in detail in next sections.

## 2.2 Quick review of gauge/gravity correspondence

In this brief review, we outline the core aspects of the AdS/CFT paradigm, focusing on its foundational framework. The discussion begins with the *AdS/CFT duality*, often referred to as the GKPW dictionary [8, 9, 75], which asserts an equivalence between the partition functions of two distinct theories

$$\mathcal{Z}_{\mathrm{bdry}}\left[\mathcal{J}^i(x)\right] = \mathcal{Z}_{\mathrm{bulk}}\left[\mathcal{J}^i(x)\right], \tag{2.5}$$

where

$$\mathcal{Z}_{\text{bdry}}\left[\mathcal{J}^i(x)\right] = \int D\phi(x)\, \exp\left[-S_{\text{CFT}}[\phi(x)] - \int_\Sigma \mathrm{d}^d x\, \sqrt{-\gamma}\, \mathcal{O}_i(x)\, \mathcal{J}^i(x)\right], \tag{2.6a}$$

$$\mathcal{Z}_{\text{bulk}}\left[\mathcal{J}^i(x)\right] = \int_{J^i(x,r_\infty)=r_\infty^{d-\Delta}\mathcal{J}^i(x)} DJ^i(x,r)\, \exp\left(-S_{\text{bulk}}\right), \tag{2.6b}$$

where we have adopted the coordinates where $(x^a, r)$, $a = 0, 1, \cdots, d-1$ parametrize the asymptotically AdS$_{d+1}$ spacetime, $r$ denotes its radial coordinate and $x^a$ spans the timelike asymptotic causal boundary of AdS space $\Sigma$ which is located at larger $r$, (see Fig.1). In particular, we take the metric of the asymptotic AdS space to have the standard Fefferman-Graham near boundary (large $r$):

$$\mathrm{d}s^2 = \ell^2\frac{\mathrm{d}r^2}{r^2} + r^2\gamma_{ab}\,\mathrm{d}x^a\,\mathrm{d}x^b + \cdots, \tag{2.7}$$

where $\ell$ is the AdS radius and $\cdots$ denotes terms subleading in $\ell/r$ expansion. $\phi(x)$ denotes a generic field in the $d$ dimensional CFT on $\Sigma$, $\mathcal{O}_i(x)$ is a generic gauge-invariant single-trace local operator of scaling dimension $\Delta$ in the CFT and $\mathcal{J}^i(x)$ is its coupling which has scaling dimension $d-\Delta$. $Z_{\text{bulk}}$ represents the partition function of quantum gravity in a $(d+1)$-dimensional asymptotically AdS spacetime. The bulk fields $J^i(x,r)$ are evaluated with Dirichlet boundary conditions $\delta J^i(x, r_\infty) = 0$. Therefore, the partition function $Z_{\text{bulk}}$ depends on the boundary values of the bulk fields, related to the deformations couplings of the CFT side as

$$\mathcal{J}^i(x) = r_\infty^{\Delta-d} J^i(x, r_\infty). \tag{2.8}$$

The scaling dimension of $\mathcal{O}_i(x)$ determines/is determined by the mass of the bulk field $m$. For a scalar field, $\Delta, m$ are related as $\Delta(\Delta - d) = m^2\ell^2$; conversely, given $m$, $\Delta$ is the larger root of this equation. For non-scalar fields $\Delta(\Delta - d) = m^2\ell^2 + f(s)$, where $f(s)$ is a functions of spacetime dimension $d$ and Lorentz representation of the deformation operator $\mathcal{O}_i$ [9].

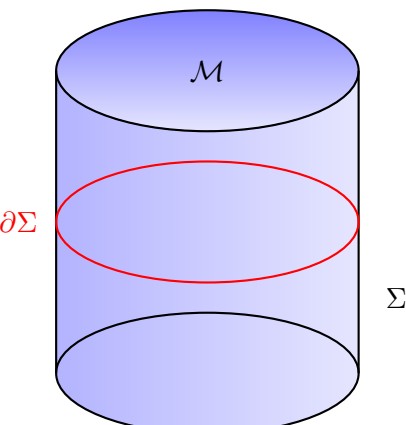

**Figure 1**: $\mathcal{M}$ denotes a $(d+1)$-dimensional asymptotically AdS spacetime and $\Sigma$ is its timelike asymptotic causal boundary. $\partial\Sigma$ denotes a $d-1$ dimensional Cauchy (constant time) slice on the AdS boundary $\Sigma$.

**Gauge/Gravity correspondence.** There is a special limit of AdS/CFT (2.5) where the bulk gravitational theory is classical when $l_{\text{Pl}}^2\mathcal{R} \ll 1$ (where $l_{\text{Pl}}$ is $d+1$ dimensional Planck length and $\mathcal{R}$ is a typical curvature radius of the background) and the boundary theory is well approximated by its planar limit (while typically strongly coupled) and has a large number of degrees of freedom, $N \gg 1$ ($N$ denotes the number of boundary degrees of freedom). We refer to this limit of the duality as *gauge/gravity*

*correspondence.* In this limit, the AdS/CFT duality (2.5) reduces to

$$S_{\text{bdry}}[\phi^*] + \int_\Sigma \mathrm{d}^d x \sqrt{-\gamma}\, \mathcal{J}^i\, \mathcal{O}_i[\phi^*] = S_{\text{bulk}}[\mathcal{J}^i(x)]\,, \tag{2.9}$$

where $\phi^*$ is the solution to the following saddle point equation

$$\frac{\delta S_{\text{bdry}}[\phi^*]}{\delta\phi} + \int_\Sigma \mathrm{d}^d x \sqrt{-\gamma}\, \mathcal{J}^i \frac{\delta \mathcal{O}_i[\phi^*]}{\delta\phi} = 0\,, \quad \Longrightarrow \quad \phi^* = \phi^*[\mathcal{J}^i]\,. \tag{2.10}$$

This equation simply shows that the LHS of (2.9) is a function of $\mathcal{J}^i$. For later use, we also rewrite this equation as follows

$$\frac{\delta S_{\text{bdry}}[\phi^*]}{\delta\phi} + \int_\Sigma \mathrm{d}^d x \sqrt{-\gamma}\, r_\infty^{d-\Delta} J^i(x, r_\infty) \frac{\delta \mathcal{O}_i[\phi^*]}{\delta\phi} = 0\,. \tag{2.11}$$

We close this very quick review of AdS/CFT with some comments [9]:

1. To elucidate the meaning of $\phi^*$, consider the expectation value of deformation operator $\mathcal{O}_i$,

   $$\langle \mathcal{O}_i \rangle_{\mathcal{J}} = -\frac{\delta \ln \mathcal{Z}_{\text{bdry}}}{\delta \mathcal{J}^i} = \mathcal{O}_i[\phi^*[\mathcal{J}^i]]\,, \tag{2.12}$$

   where the last two equalities are written in the saddle point approximation. To simplify the notation and whenever there is no confusion, we will denote the value of $\mathcal{O}_i$ at the saddle point simply by $\mathcal{O}_i$, that is, $\mathcal{O}_i[\phi^*[\mathcal{J}^i]] = \langle \mathcal{O}_i \rangle_{\mathcal{J}} = \mathcal{O}_i$.

2. Fields in the gravity side of the correspondence (2.9), $J_i$, generically satisfy differential equation on the $\text{AdS}_{d+1}$ space that is second order in AdS radial coordinate $r$. There are typically two solutions one of which is regular in the interior ($r \ll \ell$, $\ell$ being the AdS radius) and goes like $r^{d-\Delta}$ near the AdS boundary and the other which has a singularity in the interior. We choose the regular solution.

3. Besides the regularity, we should specify the boundary condition, the value of the field at a given $r$, which is typically chosen to be $r_\infty$. With these choices, one can then uniquely specify a gravity solution through (2.8) and the field theory coupling of operators $\mathcal{O}_i(x)$, $\mathcal{J}_i(x)$.

4. In summary, general solutions on the gravity side depend on two sets of codimension-one data: $J^i(x, r_\infty)$ and $O_i(x, r_\infty)$. These quantities are canonically conjugate and are related to quantities $\mathcal{J}^i$ and $\mathcal{O}_i$ on the CFT side up to appropriate scaling with $r_\infty$, $J^i(x, r_\infty) = r_\infty^{\Delta-d} \mathcal{J}^i(x), O_i(x, r_\infty) = r_\infty^{-\Delta} \mathcal{O}_i[\mathcal{J}(x)]$. More explicitly, $J_i(x, r)$ as a regular bulk solution admits the near boundary expansion,

   $$J_i(x, r) = r^{\Delta-d} \mathcal{J}^i(x) + r^{d-\Delta-1} \mathcal{J}_{1i}[\mathcal{J}_i] + \cdots + r^{-\Delta} \mathcal{O}_i[\mathcal{J}(x)] + \cdots\,. \tag{2.13}$$

   Note that the coefficient of $r^{-\Delta}$ term, through the regularity condition, is now a functional of the leading term $\mathcal{J}_i$. See the appendix for an explicit calculation for a free scalar field.

5. A canonical choice for boundary conditions of the gravity fields is the Dirichlet boundary condition. To impose this boundary condition appropriate boundary terms should be added to the bulk (super)gravity action. For instance, in the case of Einstein's gravity, the well-known Gibbons-Hawking-York term [10, 11] serves this purpose. Such boundary terms should be included in (2.9) and in (2.10) in finding and evaluating the saddle point.

6. To ensure that the on-shell action in (2.9) is finite, extra boundary Lagrangians are required to regulate any divergences. These should not change the prescribed boundary condition. For the remainder of this discussion, we will assume that the bulk action incorporates all the necessary boundary terms. More precisely, we have

$$\delta S_{\text{bulk}}\Big|_{\text{on-shell}} = \int_{\Sigma} \mathrm{d}^d x \sqrt{-\gamma}\, r_{\infty}^d\, O_i(x, r_{\infty})\, \delta J^i(x, r_{\infty}) , \tag{2.14}$$

which ensures that the bulk theory has a well-defined action principle with the Dirichlet boundary condition $\delta J^i(x, r_{\infty}) = 0$.

# 3 Gauge/gravity correspondence and W-freedom

As discussed in the previous sections, it is quite natural to revisit the gauge/gravity correspondence from the CPSF viewpoint, and in particular to understand the role of the $\mathbf{W}$, $\mathbf{Y}$, and $\mathbf{Z}$ freedoms in the correspondence. In this section, we consider the role of the $\mathbf{W}$-freedom in the gauge/gravity correspondence, revisit the $\mathbf{W}$-freedom on the bulk side in gauge/gravity limit, and then explore its counterpart in the dual boundary theory. However, before proceeding, it is useful to begin with a "dimensional analysis".

**Dimensional analysis of $(p, q)$-forms.** The AdS/CFT duality (or its large $N$ limit, known as the gauge/gravity correspondence) identifies the solution phase spaces on both sides of the duality. Specifically, the $q$-form component of the $(p, q)$-forms in the bulk and boundary are defined on the same solution phase space. The following differential structures can be recognized in both the bulk and boundary theories.

- On the bulk side, we consider the Lagrangian $\mathbf{L}_{\text{bulk}}$, the symplectic potential $\mathbf{\Theta}_{\text{bulk}}$, and the freedoms $\mathbf{W}_{\text{bulk}}, \mathbf{Y}_{\text{bulk}}$, and $\mathbf{Z}_{\text{bulk}}$, which are respectively $(d + 1, 0)$, $(d, 1)$, $(d, 0)$, $(d - 1, 1)$, and $(d - 1, 0)$ forms in an (asymptotic) AdS$_{d+1}$ spacetime.

- On the boundary side, we have the Lagrangian of the dual field theory $\mathbf{L}_{\text{bdry}}$, its symplectic potential $\mathbf{\Theta}_{\text{bdry}}$, and the freedoms $\mathbf{W}_{\text{bdry}}, \mathbf{Y}_{\text{bdry}}$, and $\mathbf{Z}_{\text{bdry}}$, which are respectively $(d, 0)$, $(d - 1, 1)$, $(d - 1, 0)$, $(d - 2, 1)$, and $(d - 2, 0)$ forms on $\Sigma$, the $d$-dimensional causal boundary of AdS$_{d+1}$.

Therefore,

A. $\mathbf{L}_{\text{bdry}}$ and $\mathbf{W}_{\text{bulk}}$ are both $(d, 0)$-forms;

B. $\mathbf{W}_{\text{bdry}}$ and $\mathbf{Z}_{\text{bulk}}$ have the same degree, they are both $(d - 1, 0)$-forms;

C. $\mathbf{\Theta}_{\text{bdry}}$ and $\mathbf{Y}_{\text{bulk}}$ are both $(d - 1, 1)$-forms.

The forms $\mathbf{Y}_{\text{bdry}}$ and $\mathbf{Z}_{\text{bdry}}$, which are $(d - 2)$-forms on the spacetime, contribute to the duality analysis only when integrated over $(d - 2)$-dimensional sectors of the spacetime. However, since we assume that $\partial \Sigma$ is compact and has no boundary, $\mathbf{Y}_{\text{bdry}}$ and $\mathbf{Z}_{\text{bdry}}$ do not play a role in the AdS/CFT analysis. Thus, we may set them to zero.

In this section, we discuss how the quantities introduced in item A above are related in our analysis of the AdS/CFT correspondence. In the next section, we examine the quantities mentioned in items B and C.

**Notation:** In the following, we will occasionally use the Hodge duals of the differential forms introduced in this section and raise their indices using the metric. In doing so, we will use the same notation but omit the boldface. For example, we will consider quantities such as $W_{\text{bulk}}^{\mu}$, $Y_{\text{bulk}}^{\mu\nu}$, etc., which are derived from $\mathbf{W}_{\text{bulk}}$ and $\mathbf{Y}_{\text{bulk}}$. When working in AdS space, the relevant component for our analysis is $W_{\text{bulk}}^{r}$, which will be simply denoted by $W_{\text{bulk}}$.

## 3.1 W-freedom, bulk view

Consider a bulk theory, the solution phase space of which is specified by canonically conjugate functions $O_i, J^i$:

$$S_{\text{bulk}} = S_{\text{bulk}}[O_i, J^i], \qquad \delta S_{\text{bulk}}[O_i, J^i] \stackrel{\circ}{=} \int_{\Sigma} \mathrm{d}^d x \, O_i \, \delta J^i. \tag{3.1}$$

One may span the same solution phase space in a different slicing by performing a canonical transformation, in terms of two other canonically conjugate variables $\bar{O}_i, \bar{J}^i$:

$$\bar{S}_{\text{bulk}} = \bar{S}_{\text{bulk}}[\bar{O}_i, \bar{J}^i], \qquad \delta \bar{S}_{\text{bulk}}[\bar{O}_i, \bar{J}^i] \stackrel{\circ}{=} \int_{\Sigma} \mathrm{d}^d x \, \bar{O}_i \, \delta \bar{J}^i, \tag{3.2}$$

where

$$\bar{S}_{\text{bulk}}[\bar{O}_i, \bar{J}^i] = S_{\text{bulk}}[O_i, J^i] + \int_{\Sigma} \mathrm{d}^d x \, W_{\text{bulk}}[O_i, J^i],$$
$$\bar{O}_i = \bar{O}_i[O_i, J^i], \qquad \bar{J}^i = \bar{J}^i[O_i, J^i]. \tag{3.3}$$

That is, the two Lagrangians should differ by a boundary term (so that they lead to the same equations of motion). Note that in both of barred and unbarred slicing, the variational principle is fulfilled by the usual Dirichlet boundary conditions, i.e. $\delta J^i = 0$ or $\delta \bar{J}^i = 0$.

The above three equations then yield

$$O_i + \frac{\delta W_{\text{bulk}}}{\delta J^i} = \bar{O}_j \frac{\delta \bar{J}^j}{\delta J^i}, \qquad \frac{\delta W_{\text{bulk}}}{\delta O_i} = \bar{O}_j \frac{\delta \bar{J}^j}{\delta O_i}. \tag{3.4}$$

These equations serve as equations for determining $\bar{O}_i$ and $\bar{J}^i$. In the standard terminology, $W_{\text{bulk}}$ is the generator of canonical transformations/change of slicing that relates bar and unbar slicings.

The above discussion treated the conjugate variables $O_i, J^i$ as independent. However, as discussed in the AdS/CFT setting regularity of the bulk solutions specifies $O_i = O_i[J^i]$. Restricting to these regular solutions, the on-shell variation of the action becomes a total variation:

$$O_i \, \delta J^i \stackrel{\circ}{=} \delta G[J^i], \qquad O_i[J^i] = \frac{\delta G}{\delta J^i}. \tag{3.5}$$

In a similar manner, for the barred-slice we have $\bar{O}_i[\bar{J}^i] = \frac{\delta \bar{G}}{\delta \bar{J}^i}$ and

$$\bar{G}[\bar{J}^i] = G[J^i] + W_{\text{bulk}}[J^i, O_i[J^i]]. \tag{3.6}$$

In the gauge/gravity correspondence $G[J^i]$ is the generating function of the boundary theory in the saddle point approximation. The equation above demonstrates that the addition of the $W$-term to the bulk action introduces a deformation in the boundary theory. Below, we will explore this point further.

## 3.2 Bulk W-freedom corresponds to multi-trace deformation of the boundary theory

In [27], Witten proposes a procedure to incorporate multi-trace deformations in quantum field theory by using a general boundary condition for the bulk fields. A similar proposal appeared around the same time in [29, 31]. The proposal was further explored in [28, 76], developed for double-trace deformations in [41, 42], generalized for regular and irregular boundary conditions in [30], and extended to vector gauge fields in the bulk in [77]. In our framework, (3.6) is the key to understanding what corresponds to $W$-freedom on the boundary theory side of the correspondence: *W-freedom is associated with introducing a deformation in the boundary theory.* Here we provide a derivation of Witten's proposal. To this end, we start with (3.1) and rewrite $J^i, O_i$ in terms of their boundary field theory counterparts $\mathcal{J}^i, \mathcal{O}_i$. Recall (2.9) and the saddle-point equation (2.10)

$$S_{\text{bdry}}[\phi^*[\mathcal{J}^i]] + \int_\Sigma \mathrm{d}^d x \sqrt{-\gamma}\, \mathcal{J}^i\, \mathcal{O}_i[\phi^*[\mathcal{J}^i]] = S_{\text{bulk}}[\mathcal{J}^i(x)]\,, \tag{3.7}$$

and add an arbitrary "new deformation" term $\int_\Sigma \mathrm{d}^d x \sqrt{-\gamma}\, \mathcal{W}[\mathcal{O}_i[\mathcal{J}^i], \mathcal{J}^i]$ to both sides of the equation:

$$S_{\text{bdry}}[\phi^*[\mathcal{J}^i]] + \int_\Sigma \mathrm{d}^d x \sqrt{-\gamma}\, \mathcal{W}[\mathcal{O}_i[\mathcal{J}^i], \mathcal{J}^i] = S_{\text{bulk}}[\mathcal{J}^i] + \int_\Sigma \mathrm{d}^d x \sqrt{-\gamma}\, \left(\mathcal{W}[\mathcal{O}_i[\mathcal{J}^i], \mathcal{J}^i] - \mathcal{J}^i\, \mathcal{O}_i[\mathcal{J}^i]\right)\,. \tag{3.8}$$

The above equation may be written in a more suggestive form

$$\bar{S}_{\text{bdry}}[\phi^*[\mathcal{J}^i]] = \bar{S}_{\text{bulk}}[\mathcal{J}^i]\,, \tag{3.9}$$

where

$$\bar{S}_{\text{bdry}}[\phi] = S_{\text{bdry}}[\phi] + \int_\Sigma \mathrm{d}^d x \sqrt{-\gamma}\, \mathcal{W}[\mathcal{O}_i[\phi], \mathcal{J}^i]\,, \tag{3.10a}$$

$$\bar{S}_{\text{bulk}}[\mathcal{J}^i] = S_{\text{bulk}}[\mathcal{J}^i] + \int_\Sigma \mathrm{d}^d x\, W_{\text{bulk}}[\mathcal{O}_i, \mathcal{J}^i]\,, \tag{3.10b}$$

with

$$\boxed{W_{\text{bulk}}[\mathcal{O}_i, \mathcal{J}^i] = \sqrt{-\gamma}\left(\mathcal{W}[\mathcal{O}_i, \mathcal{J}^i] - \mathcal{J}^i\, \mathcal{O}_i\right)\,.} \tag{3.11}$$

Let us discuss the above equations. We first note that (3.9) may be viewed as the extension of the original correspondence equation (2.9) for deformed theories that importantly, by construction, has the same saddle point configuration $\phi^*[\mathcal{J}^i]$. If $\mathcal{O}_i$ is a single-trace operator, a generic functional $\mathcal{W}[\mathcal{O}_i[\phi], \mathcal{J}^i]$ is generically a multi-trace deformation, i.e. (3.10a) is a multi-trace deformation of the original boundary theory. Likewise, (3.10b) defines a bulk theory with boundary term $W$ (3.11). As discussed, this is a bulk theory with a modified boundary condition.

Deforming the boundary theory with multi-trace deformation $\mathcal{W}[\mathcal{O}_i]$, $\bar{S}_{\text{bdry}}$ will be a function of $\mathcal{W}$ and one should still define the dual variable $\mathcal{J}^i$. To this end, one should consider the variational principle for the bulk gravitational theory with $W$-freedom. The variation of the bulk action is given by

$$\delta \bar{S}_{\text{bulk}} = \int_\Sigma \mathrm{d}^d x \left[-\left(\mathcal{J}^i - \frac{\delta(\sqrt{-\gamma}\, \mathcal{W})}{\delta(\sqrt{-\gamma}\, \mathcal{O}_i)}\right)\delta(\sqrt{-\gamma}\, \mathcal{O}_i) + \frac{\delta(\sqrt{-\gamma}\, \mathcal{W})}{\delta\mathcal{J}^i}\delta\mathcal{J}^i\right]\,. \tag{3.12}$$

To ensure a well-defined variational principle with Dirichlet boundary condition in bulk, as discussed in the previous subsection, one should impose the following boundary condition

$$\mathcal{J}^i = \frac{\delta(\sqrt{-\gamma}\, \mathcal{W})}{\delta(\sqrt{-\gamma}\, \mathcal{O}_i)}\,. \tag{3.13}$$

We hence recover/derive Witten's proposal for gravitational boundary conditions in the presence of multi-trace deformations of the boundary theory [27].

Having stated our main result, we would like to stress an important fact. Our analysis above is consistent with the assumption that the saddle-point configuration of the boundary field theory $\phi^*[\mathcal{J}^i]$ remains unchanged before and after the deformation. At first glance, this may seem puzzling as (3.10a) explicitly shows the actions of the boundary theory before and after the deformation are different. Therefore, equations of motion (saddle-point equations) are also different. However, the crucial point is that the sources/couplings $\mathcal{J}^i$ are also modified by the deformation. Once the deformation is introduced, the source must be adjusted accordingly, as dictated by (3.13). In other words, (3.13) acts as the consistency condition that ensures this correspondence.

The above analysis was performed in the saddle point approximation of the large $N$ gauge/gravity correspondence. One may hence conjecturally extend the above to the full AdS/CFT duality setup with multi-trace deformations:

$$\left\langle \exp\left( \int_\Sigma \mathrm{d}^d x \sqrt{-\gamma}\, \mathcal{W}[\mathcal{O}, \mathcal{J}^i] \right) \right\rangle_{\mathrm{CFT}} = \bar{\mathcal{Z}}_{\mathrm{bulk}}[\mathcal{J}^i] \,, \tag{3.14}$$

where $\mathcal{O}_i, \mathcal{J}^i$ are related as in (3.13).

## 4 Gauge/gravity correspondence and $\mathbf{Y}, \mathbf{Z}$ freedoms

The gauge/gravity correspondence can be written as equality of on-shell actions of bulk and boundary parts. Given the action, CPSF determines on-shell symplectic potential up to $\mathbf{W}, \mathbf{Y}, \mathbf{Z}$ freedoms (2.2). In the previous section, we explored the role and physical meaning of $\mathbf{W}_{\mathrm{bulk}}$ freedom. In this section we explore the $\mathbf{Y}_{\mathrm{bulk}}$ and $\mathbf{Z}_{\mathrm{bulk}}$ freedoms and show that, as anticipated from the discussions in the opening of Section 3, the former is equal to the symplectic potential of the boundary theory and the latter corresponds to the $\mathbf{W}_{\mathrm{bdry}}$ freedom.

To establish the expected proposal made above, consider variation of the AdS/CFT duality (2.5)

$$\delta \mathcal{Z}_{\mathrm{bdry}} = \delta \mathcal{Z}_{\mathrm{bulk}} \,, \tag{4.1}$$

which upon anomaly-free conditions $\delta D\phi = 0$ and $\delta DJ^i = 0$, can be explicitly written as

$$\delta \mathcal{Z}_{\mathrm{bdry}} = \int D\phi \, \exp\left[ -S_{\mathrm{bdry}} - \int_\Sigma \sqrt{-\gamma}\, \mathcal{J}^i \mathcal{O}_i \right] \left( \delta S_{\mathrm{bdry}} + \int_\Sigma \sqrt{-\gamma}\, \mathcal{J}^i \delta \mathcal{O}_i \right) ,$$
$$\delta \mathcal{Z}_{\mathrm{bulk}} = \int_{J^i(x,r_\infty)=r_\infty^{d-\Delta}\mathcal{J}^i(x)} DJ^i(x,r) \, \exp\left( -S_{\mathrm{bulk}} \right) \delta S_{\mathrm{bulk}} \,. \tag{4.2}$$

In the large $N$ limit the above yields

$$\delta S_{\mathrm{bdry}}[\phi^*] + \int_\Sigma \sqrt{-\gamma}\, \mathcal{J}^i \delta \mathcal{O}_i(\phi^*) = \delta S_{\mathrm{bulk}}[J^{i*}] \,, \tag{4.3}$$

where $\phi^*, J^{i*}$ respectively denote the solution to the boundary and bulk equations of motion. We note that in the saddle point, we have

$$\delta S_{\mathrm{bdry}}[\phi^*] + \int_\Sigma \sqrt{-\gamma}\, \mathcal{J}^i \delta \mathcal{O}_i(\phi^*) \stackrel{\circ}{=} \int_\Sigma \mathrm{d}\boldsymbol{\Theta}_{\mathrm{bdry}} \,, \qquad \delta S_{\mathrm{bulk}}[J^{i*}] \stackrel{\circ}{=} \int_\mathcal{M} \mathrm{d}\boldsymbol{\Theta}_{\mathrm{bulk}}^{\mathrm{D}} \,. \tag{4.4}$$

Consequently, we get the following interesting equation

$$\int_{\Sigma} \mathrm{d}\boldsymbol{\Theta}_{\mathrm{bdry}} \overset{\circ}{=} \int_{\mathcal{M}} \mathrm{d}\boldsymbol{\Theta}^{\mathrm{D}}_{\mathrm{bulk}}, \qquad \Longrightarrow \qquad \boldsymbol{\Omega}_{\mathrm{bdry}} \overset{\circ}{=} \boldsymbol{\Omega}_{\mathrm{bulk}}. \tag{4.5}$$

That is, as implied by or expected from the correspondence or duality, the on-shell symplectic forms of the boundary and bulk theories should be equal.

Next, we take the integral of both sides of (4.5) using the Stokes theorem. Let the asymptotic AdS space $\mathcal{M}$ have only one asymptotic boundary $\Sigma$ and the "bulk solution space" consist of field configurations that are *regular* inside $\mathcal{M}$. With these assumptions,[1]

$$\Delta \int_{\partial\Sigma} \boldsymbol{\Theta}_{\mathrm{bdry}} = \int_{\Sigma} \boldsymbol{\Theta}^{\mathrm{D}}_{\mathrm{bulk}}, \tag{4.6}$$

where $\Delta X(\partial\Sigma) := X(\partial\Sigma_{+\infty}) - X(\partial\Sigma_{-\infty})$ and to simplify the notation and wherever there is no risk of confusion, in (4.6) and hereafter the standard equality sign $=$ should be viewed as on-shell equality $\overset{\circ}{=}$.

We now demonstrate how (4.5) and (4.6) can be used to fix $\mathbf{Y}_{\mathrm{bulk}}$ and $\mathbf{Z}_{\mathrm{bulk}}$. To this end, we recall that in the previous sections, we fixed $W$ by imposing Dirichlet boundary conditions. In the following, we assume Dirichlet boundary conditions once again and fix the remaining degrees of freedom. Therefore, according to (2.2), we have

$$\boldsymbol{\Theta}^{\mathrm{D}}_{\mathrm{bulk}} = \bar{\boldsymbol{\Theta}}^{\mathrm{D}}_{\mathrm{bulk}} + \mathrm{d}\mathbf{Y}_{\mathrm{bulk}} + \mathrm{d}\delta\mathbf{Z}_{\mathrm{bulk}}, \tag{4.7}$$

where $\bar{\boldsymbol{\Theta}}^{\mathrm{D}}_{\mathrm{bulk}}$ includes the $\mathbf{W}_{\mathrm{bulk}}$ and is chosen such that it guarantees Dirichlet boundary conditions fo the bulk fields; $\mathbf{W}$ is fixed by the discussions of previous section. Explicitly,

$$\int_{\Sigma} \bar{\boldsymbol{\Theta}}^{\mathrm{D}}_{\mathrm{bulk}} = 0. \tag{4.8}$$

Plugging (4.7) into (4.6)

$$\int_{\Sigma} \bar{\boldsymbol{\Theta}}^{\mathrm{D}}_{\mathrm{bulk}} = \Delta \int_{\partial\Sigma} \left( \boldsymbol{\Theta}_{\mathrm{bdry}} - \delta\mathbf{Z}_{\mathrm{bulk}} - \mathbf{Y}_{\mathrm{bulk}} \right), \tag{4.9}$$

and using (4.8), we finally get

$$\boxed{\mathbf{Y}_{\mathrm{bulk}} = \boldsymbol{\Theta}_{\mathrm{bdry}} - \delta\mathbf{Z}_{\mathrm{bulk}}.} \tag{4.10}$$

Recalling that $\partial\Sigma$ is taken to be boundary-less, $\mathbf{Y}_{\mathrm{bdry}}, \mathbf{Z}_{\mathrm{bdry}}$ may be set to zero as they do not appear in (4.6) and hence one can expand $\boldsymbol{\Theta}_{\mathrm{bdry}}$ as

$$\boldsymbol{\Theta}_{\mathrm{bdry}} = \boldsymbol{\Theta}^{\mathrm{D}}_{\mathrm{bdry}} + \delta\mathbf{W}_{\mathrm{bdry}}, \tag{4.11}$$

where $\boldsymbol{\Theta}^{\mathrm{D}}_{\mathrm{bdry}}$ is part of on-shell boundary symplectic potential guaranteeing the Dirichlet boundary which only involves genuine propagating modes of the boundary theory, i.e. for the $\mathcal{N} = 4, D = 4$ supersymmetric $U(N)$ Yang-Mills theory these are the corresponding $\mathcal{N} = 4$ gauge multiples. One can therefore conveniently choose

$$\boxed{\mathbf{Z}_{\mathrm{bulk}} = \mathbf{W}_{\mathrm{bdry}}, \qquad \mathbf{Y}_{\mathrm{bulk}} = \boldsymbol{\Theta}^{\mathrm{D}}_{\mathrm{bdry}}.} \tag{4.12}$$

To summarize the last two sections, all six freedoms of the boundary and bulk dual theories have been fixed as shown in Table 1. In Appendix A we have presented the above procedure for fixing all the freedoms in a simple example of a massive free scalar theory in an AdS background.

---

[1]It may happen that $\mathcal{M}$ has an additional boundary besides $\Sigma$. For instance, in the case of an asymptotically AdS black hole, the black hole horizon $\mathcal{H}$ can be regarded as another boundary. In such cases, one should include the term $-\int_{\mathcal{H}} \bar{\boldsymbol{\Theta}}^{\mathrm{D}}_{\mathrm{bulk}}$ on the right-hand side of (4.6). In these cases, this term may be set to zero by imposing appropriate boundary conditions on $\mathcal{H}$.

# 5 Setting boundary conditions free

Reviewing gauge/gravity correspondence in light of CPSF provides the appropriate framework to discuss extensions of the correspondence beyond the standard Dirichlet boundary conditions. In this section and building upon analysis of Section 3, we show how the choice of $W$-freedom yields different boundary conditions: Dirichlet, Neumann, and conformal as well as the new "conformal conjugate" boundary conditions. For simplicity, we only consider pure gravity without matter fields in general spacetime dimensions. The generalization to capture matter fields is straightforward.

## 5.1 Dirichlet boundary condition

We begin with the Einstein-Hilbert action in the presence of cosmological constant with the standard Gibbons-Hawking-York boundary term

$$S = \frac{1}{16\pi G} \int d^{d+1}x \sqrt{-g}\,(R - 2\Lambda) + \frac{1}{8\pi G} \int_\Sigma d^d x \sqrt{-h}\, K + \frac{1}{8\pi G} \int_\Sigma d^d x\, \mathcal{L}_{\rm ct}\,, \tag{5.1}$$

where $\mathcal{L}_{\rm ct}$ denotes the counterterm Lagrangian, which ensures the finiteness of the on-shell action and canonical variables. We impose the condition that this counterterm does not alter the Dirichlet boundary conditions. To satisfy this, we choose the counterterm Lagrangian to be of the form $\mathcal{L}_{\rm ct}[h_{ab}, \hat{R}_{abcd}]$, where $\hat{R}_{abcd}$ is the Riemann tensor of $h_{ab}$. The variation of this Lagrangian is then given by the following expression

$$\delta \mathcal{L}_{\rm ct} = -4\pi G \sqrt{-h}\, \mathcal{T}_{\rm ct}^{ab}\, \delta h_{ab} + \partial_a \Theta_{\rm ct}^a\,. \tag{5.2}$$

We take the following parametrization for the metric

$$ds^2 = g_{\mu\nu}\, dx^\mu\, dx^\nu = N^2\, dr^2 + h_{ab}(dx^a + U^a\, dr)(dx^b + U^b\, dr)\,. \tag{5.3}$$

In comparison with (2.7), $h_{ab} = r^2 \gamma_{ab} + \mathcal{O}(r)$.

The Dirichlet symplectic potential is then given by

$$\boldsymbol{\Theta}_{\rm D} = \int_\Sigma d^d x \left\{ -\frac{1}{2}\sqrt{-h}\, \mathcal{T}^{ab} \delta h_{ab} + \frac{1}{8\pi G} \partial_a \left( \sqrt{-g}\, s^{[r} \delta s^{a]} + \Theta_{\rm ct}^a \right) \right\}\,, \tag{5.4}$$

where $s_\mu$ is the normalized normal vector on $\Sigma$ and $\mathcal{T}^{ab}$ is the renormalized Brown-York energy-momentum tensor (rBY-EMT), which is defined in terms of extrinsic curvature,

$$K_{ab} := \frac{1}{2} h_a^\alpha h_b^\beta \mathcal{L}_s h_{\alpha\beta}\,, \qquad h_{\mu\nu} := g_{\mu\nu} - s_\mu s_\nu\,, \qquad h_a^\mu = \delta_a^\mu - s^\mu s_a\,, \tag{5.5}$$

as follows [2]

$$\mathcal{T}^{ab} = \mathring{\mathcal{T}}^{ab} + \mathcal{T}_{\rm ct}^{ab}\,, \tag{5.6}$$

where

$$\mathring{\mathcal{T}}^{ab} = \frac{1}{8\pi G} \left( K^{ab} - K\, h^{ab} \right)\,, \qquad \mathring{\mathcal{T}} = h_{ab} \mathring{\mathcal{T}}^{ab} = \frac{1-d}{8\pi G} K = \frac{1-d}{8\pi G} \mathcal{L}_s(\ln \sqrt{-h})\,. \tag{5.7}$$

From the definition in (5.7), it follows that $\mathring{\mathcal{T}}^{ab}$ is divergence-free, i.e., $\nabla_a \mathring{\mathcal{T}}^{ab} = 0$, where $\nabla_a$ is the covariant derivative with respect to the boundary metric $h_{ab}$. We will show that the counter energy-momentum tensor is conserved off-shell, i.e., $\nabla_a \mathcal{T}_{\rm ct}^{ab} = 0$. Consequently, we have $\nabla_a \mathcal{T}^{ab} = 0$. In all our analyses hereafter we use the rBY-EMT $\mathcal{T}^{ab}$.

---

[2] In the standard terminology, $\mathring{\mathcal{T}}^{ab}$ is referred to as the Brown-York energy-momentum tensor, while $\mathcal{T}^{ab}$ is known as the holographic or renormalized Brown-York tensor (rBY-EMT).

Accordingly, the total symplectic potential is given by (2.2)

$$\mathbf{\Theta} = \int_\Sigma \mathrm{d}^d x \left\{ -\frac{1}{2}\sqrt{-h}\,\mathcal{T}^{ab}\delta h_{ab} + \delta W^r_{\text{bulk}} + \partial_a \left( \frac{\sqrt{-g}}{8\pi G}\,s^{[r}\delta s^{a]} + \frac{\Theta^a_{\text{ct}}}{8\pi G} + Y^{ra}_{\text{bulk}} \right) \right\}, \tag{5.8}$$

By choosing the $W$ and $Y$ freedoms as [78]

$$W^\mu_{\text{bulk}} = 0\,, \qquad Y^{\mu\nu}_{\text{bulk}} = -\frac{\sqrt{-g}}{8\pi G}\,s^{[\mu}\delta s^{\nu]} - \frac{N}{4\pi G}s^{[\mu}\Theta^{\nu]}_{\text{ct}}\,, \tag{5.9}$$

with $\Theta^r_{\text{ct}} = 0$, we find the following symplectic potential

$$\mathbf{\Theta}_{\text{Dirichlet}} = -\frac{1}{2}\int_\Sigma \mathrm{d}^d x\,\sqrt{-h}\,\mathcal{T}^{ab}\delta h_{ab}\,, \tag{5.10}$$

which guarantees the Dirichlet boundary conditions $\delta h_{ab} = 0$ at the boundary. The condition $W^\mu_{\text{bulk}} = 0$ is by definition and design (cf. (2.2) and the discussions below it). In particular, (5.9) arises from the addition of the Gibbons-Hawking-York boundary term [10, 11] to the bulk Lagrangian in (5.1), to guarantee Dirichlet boundary conditions for the metric. Note that by default the $Y$ term does not affect boundary conditions on the fields.

We conclude this subsection with a few remarks on the counterterm Lagrangian and its corresponding counter energy-momentum tensor. The explicit form of the counterterm Lagrangian depends on the spacetime dimension, with a few examples given in [19]

$$d = 2: \qquad \mathcal{L}_{\text{ct}} = -\frac{1}{\ell}\sqrt{-h}\,,$$

$$d = 3: \qquad \mathcal{L}_{\text{ct}} = -\frac{2}{\ell}\sqrt{-h}\left(1 + \frac{\ell^2}{4}\hat{R}\right)\,, \tag{5.11}$$

$$d = 4: \qquad \mathcal{L}_{\text{ct}} = -\frac{3}{\ell}\sqrt{-h}\left(1 + \frac{\ell^2}{12}\hat{R}\right)\,,$$

where $\hat{R}$ is the Ricci scalar associated with the boundary metric $h_{ab}$, and the cosmological constant is given by $\Lambda = -\frac{d(d-1)}{2\ell^2}$. The corresponding counterterm energy-momentum tensor for different spacetime dimensions is

$$d = 2: \qquad \mathcal{T}^{ab}_{\text{ct}} = \frac{1}{8\pi G\,\ell}h^{ab}\,,$$

$$d = 3: \qquad \mathcal{T}^{ab}_{\text{ct}} = \frac{1}{8\pi G\,\ell}\left(2h^{ab} - \ell^2\hat{G}^{ab}\right)\,, \tag{5.12}$$

$$d = 4: \qquad \mathcal{T}^{ab}_{\text{ct}} = \frac{1}{8\pi G\,\ell}\left(3h^{ab} - \frac{\ell^2}{2}\hat{G}^{ab}\right)\,,$$

where $\hat{G}_{ab}$ is the Einstein tensor associated with $h_{ab}$. As mentioned earlier, the counterterm energy-momentum tensor is conserved off-shell, satisfying $\nabla_a \mathcal{T}^{ab}_{\text{ct}} = 0$.

## 5.2 Neumann boundary condition

In the previous subsection, we selected the $W$-freedom to ensure a well-defined action principle under Dirichlet boundary conditions. Here, we explore the same question for Neumann boundary conditions [33, 79–81]. Neumann boundary conditions correspond to fixing the canonical momentum, which in this context is given by the rBY-EMT, explicitly

$$\delta(\sqrt{-h}\,\mathcal{T}^{ab}) = 0 \qquad \text{at the boundary.} \tag{5.13}$$

One can show that the same $Y$ as in (5.9) and $W$ given by

$$W^\mu_{\text{bulk}} = \frac{1}{2}\sqrt{-g}\,\mathcal{T}\,s^\mu\,, \qquad Y^{\mu\nu}_{\text{bulk}} = -\frac{\sqrt{-g}}{8\pi G}\,s^{[\mu}\delta s^{\nu]} - \frac{N}{4\pi G}s^{[\mu}\Theta^{\nu]}_{\text{ct}}\,, \tag{5.14}$$

lead to the Neumann symplectic potential

$$\boldsymbol{\Theta}_{\text{Neumann}} = +\frac{1}{2}\int_\Sigma \mathrm{d}^d x\, h_{ab}\,\delta(\sqrt{-h}\,\mathcal{T}^{ab}) = \boldsymbol{\Theta}_{\text{Dirichlet}} + \frac{1}{2}\delta\int_\Sigma \mathrm{d}^d x\,\sqrt{-h}\,\mathcal{T}\,, \tag{5.15}$$

which guarantees the Neumann boundary conditions (5.13).

For the case of $d = 3$ (4$d$ gravity) in the Einstein-$\Lambda$ theory, the $W$ term in (5.14) takes the form

$$W_{\text{bulk}} = \frac{\sqrt{-h}}{8\pi G}\left(-K + \frac{3}{\ell} + \frac{\ell}{4}\hat{R}\right).$$

The first term precisely cancels the Gibbons-Hawking-York term in (5.1), resulting in a "total" action under Neumann boundary conditions that includes only the Einstein-Hilbert action and counterterms

$$S^{\text{N}}_{\text{total}} := S + \int_\Sigma \mathrm{d}^d x\, W_{\text{bulk}} = \frac{1}{16\pi G}\int \mathrm{d}^{d+1}x\,\sqrt{-g}\,(R - 2\Lambda) + \frac{1}{8\pi G\,\ell}\int_\Sigma \mathrm{d}^d x\,\sqrt{-h}\left(1 - \frac{\ell^2}{4}\hat{R}\right).$$

A similar observation was made in [80].

Furthermore, as expected, and as indicated by (5.15), the Neumann and Dirichlet symplectic forms differ only by a total variation. Notably, since the $Y_{\text{bulk}}$ terms remain identical in both the Neumann and Dirichlet cases, the symplectic potential of the dual boundary theory is also expected to be the same.

## 5.3 Conformal boundary condition

As the next example, we examine conformal boundary conditions in Einstein's gravity [82–90] where the boundary metric $h_{ab}$ is fixed up to a conformal factor, while the trace of the extrinsic curvature $K$ is held fixed. In our setting which is based on CPSF, it is more convenient and physically relevant to consider the trace of rBY-EMT $\mathcal{T}$ instead of $K$. Explicitly we define our conformal boundary conditions by,

$$\delta h_{ab} = \delta X\, h_{ab}\,, \qquad X = \frac{1}{d}\ln(-h)\,, \qquad \delta\mathcal{T} = 0\,. \tag{5.16}$$

In terms of the determinant-free induced boundary metric, $\mathrm{g}_{ab} = (-h)^{-\frac{1}{d}}h_{ab}$, that is $\delta\mathrm{g}_{ab} = 0$.

Before going further we pause and compare further (5.16) and what is called conformal boundary conditions in the literature [82–90]. In the latter $\delta\mathcal{T} = 0$ of (5.16) is replaced by $\delta K = 0$.[3] Since $K$ is proportional to the trace of the unrenormalized BY-EMT, $\mathring{\mathcal{T}}$, this condition is equivalent to fixing of $\mathring{\mathcal{T}}$, i.e., $\delta\mathring{\mathcal{T}} = 0$. However, in (5.16) we fix the trace of the renormalized BY-EMT (rBY-EMT), $\delta\mathcal{T} = 0$. This alternative definition has a crucial advantage: it ensures the finiteness of the on-shell action and other physical quantities. In contrast, the standard definition leads to a divergent on-shell action, necessitating regularization with counterterms that remain unknown. We note that the distinction between (5.16) and the one in the literature disappears when we deal with a flat AdS boundary, where $\hat{R}_{abcd} = 0$ as in this case $\mathcal{T}^{ab}_{\text{ct}} \propto h^{ab}$, leading to a constant $\mathcal{T}_{\text{ct}}$, and therefore $\delta\mathcal{T} = \delta\mathring{\mathcal{T}} = \delta K = 0$. Finally, we emphasize that although we are working with a different conformal boundary condition, we can recover or compare our results with the standard boundary conditions by setting $\mathcal{L}_{\text{ct}} = 0$. This choice eliminates the counterterm contribution, $\mathcal{T}^{ab}_{\text{ct}} = 0$, reducing $\mathcal{T}^{ab}$ to $\mathring{\mathcal{T}}^{ab}$.

---

[3]We also note that the condition $\delta K = 0$ is sometimes referred to as the Anderson boundary condition [83].

The choices

$$W^{\mu}_{\text{bulk}} = \frac{1}{d}\sqrt{-g}\,\mathcal{T}\,s^{\mu}\,, \qquad Y^{\mu\nu}_{\text{bulk}} = -\frac{\sqrt{-g}}{8\pi G}\,s^{[\mu}\delta s^{\nu]} - \frac{N}{4\pi G}s^{[\mu}\Theta^{\nu]}_{\text{ct}}\,, \tag{5.17}$$

fulfills the boundary conditions in (5.16) and variational principle, as with these choices,

$$\boldsymbol{\Theta}_{\text{Conformal}} = -\frac{1}{2}\int_{\Sigma}\mathrm{d}^d x\,\sqrt{-h}\left((-h)^{1/d}\,\mathcal{T}^{ab}\delta\mathrm{g}_{ab} - \frac{2}{d}\,\delta\mathcal{T}\right). \tag{5.18}$$

The above expression is compatible with conformal boundary conditions, which require fixing $\mathrm{g}_{ab}$ and $\mathcal{T}$.

We close by noting the fact that the three boundary conditions of this section, while having different $W_{\text{bulk}}$ (reassuring the boundary conditions and variational principle), share the same $Y_{\text{bulk}}$ term. Therefore, the bulk symplectic form is the same for these cases:

$$\boldsymbol{\Omega}_{\text{bulk}} = \frac{1}{2}\int_{\Sigma}\mathrm{d}^d x\,\delta h_{ab}\wedge\delta(\sqrt{-h}\,\mathcal{T}^{ab})\,. \tag{5.19}$$

## 5.4   Conformal conjugate boundary condition

As reviewed the Dirichlet and Neumann boundary conditions are "canonical conjugates" of each other, as the former requires $\delta h_{ab} = 0$ and the latter requires vanishing of variations of its canonical conjugate, $\delta(\sqrt{-h}\mathcal{T}^{ab}) = 0$, cf. (5.19). The conformal boundary case suggests decomposing $h_{ab}$ into its determinant $-h$ and its determinant-free part $\mathrm{g}_{ab}$. This naturally yields the decomposition of the conjugate energy momentum tensor into its trace $\mathcal{T}$ and its trace-free part $\mathrm{T}^{ab}$, recalling (5.18) that is,

$$\mathrm{g}_{ab} := (-h)^{-1/d}h_{ab}\,, \qquad \mathrm{T}^{ab} := (-h)^{1/d}\,\mathcal{T}^{ab} - \frac{\mathcal{T}}{d}\mathrm{g}^{ab} = (-h)^{1/d}\left(\mathcal{T}^{ab} - \frac{\mathcal{T}}{d}h^{ab}\right),$$
$$\mathrm{g}^{ac}\mathrm{g}_{cb} = \delta^{a}{}_{b}\,, \qquad \mathrm{g}_{ab}\mathrm{T}^{ab} = 0\,, \qquad \nabla^{\mathrm{g}}_{a}\mathrm{T}^{ab} = 0\,, \tag{5.20}$$

where $\nabla^{\mathrm{g}}_{a}$ is the covariant derivative w.r.t the unimodular metric $\mathrm{g}_{ab}$. With the above and starting from (5.18) and noting that $\mathrm{g}_{ab}\delta\mathrm{g}^{ab} = 0, \delta\mathrm{g}_{ab}\wedge\delta\mathrm{g}^{ab} = 0$ (both following from $\det(\mathrm{g}_{ab}) = -1$), the symplectic form may be decomposed as

$$\delta h_{ab}\wedge\delta(\sqrt{-h}\,\mathcal{T}^{ab}) = \delta\mathrm{g}_{ab}\wedge\delta(\sqrt{-h}\,\mathrm{T}^{ab}) + \frac{2}{d}\delta\sqrt{-h}\wedge\delta\mathcal{T}\,. \tag{5.21}$$

That is, canonical conjugate of $\mathrm{g}_{ab}$ and $\sqrt{-h}$ are $\sqrt{-h}\,\mathrm{T}^{ab}$ and $\frac{2}{d}\mathcal{T}$, respectively.

Eq.(5.21) provides four choices for the boundary conditions:

(I)   $\delta\mathrm{g}_{ab} = 0$ and $\delta\sqrt{-h} = 0$, recovering the Dirichlet boundary conditions discussed in section 5.1.

(II)   $\delta\mathrm{g}_{ab} = 0$ and $\delta\mathcal{T} = 0$, recovering the conformal boundary conditions discussed in section 5.3.

(III)   $\delta(\sqrt{-h}\,\mathrm{T}^{ab}) = 0$ and $\delta\sqrt{-h} = 0$ which is a new case. In this case, we require vanishing variations of the field canonically conjugate to those in the conformal boundary conditions and hence call it "conformal conjugate boundary conditions".

(IV)   $\delta(\sqrt{-h}\,\mathrm{T}^{ab}) = 0$ and $\delta\mathcal{T} = 0$, which despite similarity is not the Neumann boundary conditions discussed in section 5.2.

We start with the comment that all the above four cases and the Neumann boundary conditions in section 5.2 have the same $Y_{\text{bulk}}$ term and hence the same symplectic form as in (5.19). Let us discuss the two new cases in detail.

**Conformal conjugate boundary condition.**  For this case the variational principle is guaranteed once we choose

$$W^\mu_{\text{bulk}} = 0, \qquad Y^{\mu\nu}_{\text{bulk}} = -\frac{\sqrt{-g}}{8\pi G} s^{[\mu}\delta s^{\nu]} - \frac{N}{4\pi G} s^{[\mu}\Theta^{\nu]}_{\text{ct}}, \qquad (5.22)$$

yielding

$$\boldsymbol{\Theta}_{\text{Conf. Conj.}} = \frac{1}{2}\int_\Sigma \mathrm{d}^d x \left( \mathrm{g}_{ab}\, \delta(\sqrt{-h}\, \mathrm{T}^{ab}) - \frac{2}{d}\mathcal{T}\delta\sqrt{-h} \right). \qquad (5.23)$$

As we see $W_{\text{bulk}}$ for this case and the Dirichlet case (5.9) are the same and one can also explicitly show that (5.23) and (5.10) are equal. Technically, to show this we have used identities

$$\begin{aligned}
\mathrm{g}_{ab}\mathrm{T}^{ab} &= h_{ab}(\mathcal{T}^{ab} - \frac{1}{d}\mathcal{T}h^{ab}) = 0, \\
\mathrm{g}_{ab}\,\delta(\sqrt{-h}\mathrm{T}^{ab}) &= -\sqrt{-h}\,\mathrm{T}^{ab}\,\delta\mathrm{g}_{ab}, \\
h_{ab}\,\delta\left( \sqrt{-h}(\mathcal{T}^{ab} - \frac{1}{d}\mathcal{T}h^{ab}) \right) &= -\sqrt{-h}\,(\mathcal{T}^{ab} - \frac{1}{d}\mathcal{T}h^{ab})\,\delta h_{ab}.
\end{aligned} \qquad (5.24)$$

Despite the fact that $\boldsymbol{\Theta}_{\text{Dirichlet}} = \boldsymbol{\Theta}_{\text{Conf. Conj.}}$, the two Dirichlet and conformal conjugate cases satisfy different boundary conditions and are hence physically different.

**Case (IV)** $\delta(\sqrt{-h}\,\mathbf{T}^{ab}) = 0$ **and** $\delta\mathcal{T} = 0$ **boundary condition.**  For this case the symplectic potential is the same as $\boldsymbol{\Theta}_{\text{Conformal}}$ in (5.18), and with the same $W_{\text{bulk}}$ term as in (5.17). We note that the Neumann boundary condition case of section 5.2 is not covered in our four cases above. One may directly compare case (IV) and Neumann boundary conditions. To this end we note that $\delta(\sqrt{-h}\,\mathrm{T}^{ab}) = 0$ and $\delta\mathcal{T} = 0$ imply that

$$\delta(\sqrt{-h}\mathcal{T}^{ab}) = \frac{2}{d}(\mathcal{T}^{ab} - \frac{1}{d}\mathcal{T}h^{ab})\delta\sqrt{-h} + \frac{1}{d}\mathcal{T}\delta(\sqrt{-h}h^{ab}), \qquad \delta\mathcal{T} = 0. \qquad (5.25)$$

That is, case (IV) yields a mixture of Neumann and Dirichlet boundary conditions with field-dependent coefficients.

We close this part by the comment that as the discussions above demonstrate, the relation between boundary conditions and $W$ terms is not one-to-one; the cases (I) and (III), and (II) and (IV), have the same $W$ term while corresponding to two different boundary conditions. Explicitly, (5.24) shows that this relation is two-fold: a given $W$ can correspond to two boundary conditions.

## 6 Hydrodynamical deformations

Renormalized BY-EMT $\mathcal{T}^{ab}$ is the canonical conjugate to the boundary induced metric and in the context of holography, is the expectation value of the boundary energy-momentum tensor [14, 19]. Invariance under boundary preserving diffeomorphisms ensures that this quantity is divergence-free $\nabla_a\mathcal{T}^{ab} = 0$, where $\nabla_a$ denotes the covariant derivative compatible with $h_{ab}$. This equation is part of the constraint equations arising from the bulk equations of motion. $h_{ab}$ and $\mathcal{T}^{ab}$ are canonical conjugates, cf. (5.19), and hence, they naturally lead to a hydrodynamic interpretation of the gravitational solution space, or equivalently, of the boundary theory.

On the other hand, we have shown that $W$-freedom acts as a generator of change of slicing (canonical transformations), i.e. $W$ terms change gravitational canonical pairs while keeping the symplectic form intact. In the previous section, we discussed four cases for two of which the hydrodynamics conjugate variables are $h_{ab}$ and $\mathcal{T}^{ab}$ and for the other two they are $\mathrm{g}_{ab}, \sqrt{-h}$ and $\mathrm{T}^{ab}, \mathcal{T}$. In this section, we explore the question of whether there are other canonical pairs $\tilde{h}_{ab}, \tilde{\mathcal{T}}^{ab}$ such that $\tilde{\nabla}_a\tilde{\mathcal{T}}^{ab} = 0$. We construct a

one-function family of such *hydrodynamical pairs* and discuss the $W$ freedoms and boundary conditions associated with each member of the family. The four cases of the previous section are examples of the "hydrodynamical family". Below, we construct a one-function class in a hydrodynamical family in which $h_{ab}, \tilde{h}_{ab}$ are related by Weyl scaling and discuss the $W$ freedoms and boundary conditions associated with each member of the family. This question in the $d = 2$ case was analyzed in [78]. Our analysis here extends the cases recently discussed in [89, 90].

## 6.1 Hydrodynamics in Weyl frames

Consider two induced metrics related by a generic Weyl scaling transformation

$$h_{ab} \quad \rightarrow \quad \tilde{h}_{ab} = \Omega^{-1}\, h_{ab}\,, \tag{6.1}$$

where $\Omega$ is a general function on the spacetime and a scalar in the $\tilde{h}$-frame. To construct the divergence-free energy momentum tensor conjugate to $\tilde{h}_{ab}$, following the analysis in [78], we choose $\Omega = \Omega(\mathcal{T})$, an arbitrary function of trace of rBY-EMT $\mathcal{T}$, and define,

$$\tilde{\mathcal{T}}^{ab} := \Omega^{(1+\frac{d}{2})} \left[ \mathcal{T}^{ab} - \frac{1}{2}G(\mathcal{T})\, h^{ab} \right]\,, \qquad \tilde{\mathcal{T}} := \tilde{h}_{ab}\, \tilde{\mathcal{T}}^{ab} = \Omega^{\frac{d}{2}} \left[ \mathcal{T} - \frac{d}{2}G(\mathcal{T}) \right]\,. \tag{6.2}$$

The function $G(\mathcal{T})$ is determined in terms of $\Omega(\mathcal{T})$ via the relation

$$G' = \frac{\Omega'}{\Omega}\left( \mathcal{T} - \frac{d}{2}G \right)\,, \tag{6.3}$$

where the prime denotes differentiation with respect to the argument. For consistency, we require that $G = 0$ for a constant $\Omega$ case. A straightforward verification shows that $\tilde{\nabla}_a \tilde{\mathcal{T}}^{ab} = 0$. The divergence-free conditions: $\tilde{\nabla}_a \tilde{\mathcal{T}}^{ab} = 0$ and $\tilde{\nabla}^a \tilde{h}_{ab} = 0$ guarantee the consistency of the set of the hydrodynamic setup (and in general the gauge/gravity correspondence), see also [89].

It is evident that while $\tilde{\mathcal{T}}^{ab}$ is divergence-free, it is not trace-less. One can observe that the traceless part of $\tilde{\mathcal{T}}^{ab}$ and $\mathcal{T}^{ab}$ are proportional to each other,

$$\sqrt{-\tilde{h}}\Big( \tilde{\mathcal{T}}^{ab} - \frac{\tilde{\mathcal{T}}}{d}\tilde{h}^{ab} \Big) = \Omega\,\sqrt{-h}\Big( \mathcal{T}^{ab} - \frac{\mathcal{T}}{d}h^{ab} \Big)\,, \tag{6.4}$$

From this equation, we identify two particular combinations that remain independent of $\Omega$, i.e. they are invariants of the scaling and are the same in the original and tilde frames:

$$\tilde{\mathrm{g}}_{ab} := (-\tilde{h})^{1/d}\tilde{h}_{ab} = (-h)^{1/d}h_{ab} = \mathrm{g}_{ab}\,,$$
$$\sqrt{-\tilde{h}}\,\tilde{\mathrm{T}}^{ab} := \sqrt{-\tilde{h}}\,(-\tilde{h})^{1/d}\left( \tilde{\mathcal{T}}^{ab} - \frac{\tilde{\mathcal{T}}}{d}\tilde{h}^{ab} \right) = \sqrt{-h}\,(-h)^{1/d}\Big( \mathcal{T}^{ab} - \frac{\mathcal{T}}{d}h^{ab} \Big) = \sqrt{-h}\,\mathrm{T}^{ab}\,. \tag{6.5}$$

The above relations are expected since $\tilde{h}_{ab}$ and $h_{ab}$ are related through scaling. Furthermore, the following relation between the new and old hydrodynamic variables can be demonstrated,

$$\sqrt{-h}\,\mathcal{T}^{ab}\,\delta h_{ab} = \sqrt{-\tilde{h}}\,\tilde{\mathcal{T}}^{ab}\,\delta\tilde{h}_{ab} + \delta\left( \sqrt{-h}\,G(\mathcal{T}) \right)\,. \tag{6.6}$$

To obtain the above we used, $\delta\sqrt{-\tilde{h}} = \frac{1}{2}\sqrt{-\tilde{h}}\,\tilde{h}^{ab}\delta\tilde{h}_{ab}$. Therefore, the two original and tilde frames are related by a $W$ term, and they come with different boundary conditions on the gravity (bulk) side of the correspondence, and the associated dual (boundary) theories are related by multi-trace deformations.

**Discussions and comments on the physics of the hydrodynamic Weyl family:**

(1) For the hydrodynamical Weyl frames, the $Y$-term remains the same as in (5.9). Therefore, the symplectic form for all hydrodynamical slicings discussed here and the four cases of the previous section are the same. Furthermore, since we have $\nabla_a \mathcal{T}^{ab} = 0$ for the four cases discussed in the previous section, they can be classified within the hydrodynamical framework.

(2) Eq. (6.6) shows that for $\Omega = \Omega(\mathcal{T})$, the Weyl scaling is a canonical transformation, explicitly $\tilde{h}_{ab}$, $\sqrt{-\tilde{h}}\,\tilde{\mathcal{T}}^{ab}$ are canonical conjugates:

$$\mathbf{\Omega}_{\text{bulk}} = \frac{1}{2}\int_\Sigma \mathrm{d}^d x\, \delta h_{ab} \wedge \delta(\sqrt{-h}\,\mathcal{T}^{ab}) = \frac{1}{2}\int_\Sigma \mathrm{d}^d x\, \delta\tilde{h}_{ab} \wedge \delta\left(\sqrt{-\tilde{h}}\,\tilde{\mathcal{T}}^{ab}\right). \tag{6.7}$$

(3) Both $\mathcal{T}^{ab}$ and $\tilde{\mathcal{T}}^{ab}$ are divergence-free with respect to their respective covariant derivatives, $h_{ab}$ and $\tilde{h}_{ab}$.

(4) The function $G(\mathcal{T})$, appearing in the definition of $\tilde{\mathcal{T}}^{ab}$ (6.2), is specified in terms of $\Omega = \Omega(\mathcal{T})$. In general one can solve (6.3) to find

$$G(\mathcal{T}) = \frac{2}{d}\left(\mathcal{T} - \Omega^{-d/2}\int_0^\mathcal{T} \Omega^{d/2}\right), \tag{6.8}$$

such that $G = 0$ for constant $\Omega$. For any $\Omega$ that is a monomial in $\mathcal{T}$, $G$ is a linear function of $\mathcal{T}$. Thus, the hydrodynamic Weyl class of slicings is fully characterized by a single function of the trace of the rBY-EMT, $\mathcal{T}$.

(5) From (6.6) one learns that the $W$ term,

$$\boxed{(W_{\text{bulk}}^{\text{D}})^\mu = \frac{1}{2}\sqrt{-g}\,G(\mathcal{T})\,s^\mu\,,} \tag{6.9}$$

fixes the Dirichlet boundary conditions for the tilde-frame, $\delta\tilde{h}_{ab} = 0$. Explicitly, the tilde-frame boundary conditions in terms of the original frame variables are

$$\delta\left((-\tilde{h})^{-1/d}\tilde{h}_{ab}\right) = \delta\left((-h)^{-1/d}h_{ab}\right) = 0\,, \qquad \delta\left(\Omega^{-d}(\mathcal{T})\,h\right) = 0\,. \tag{6.10}$$

The above may be written as $d\frac{\Omega'}{\Omega}\delta\mathcal{T} = \frac{\delta h}{h}$ in which $\Omega = \Omega(\mathcal{T})$ is an arbitrary function. For $\Omega = 1$, one recovers the usual Dirichlet boundary conditions. The conformal boundary conditions $\delta\mathcal{T} = 0, \delta h \neq 0$ (5.16), can be recovered in a singular limit where $\frac{\Omega'}{\Omega}$ blows up. In this sense, the class of solutions we are considering here is an example of freelance holography, a one-function family of boundary conditions that encompasses and extends Dirichlet and conformal boundary conditions.

(6) In a similar way, the $W$ term,

$$\boxed{(W_{\text{bulk}}^{\text{N}})^\mu = \frac{1}{2}\sqrt{-g}\left[\mathcal{T} + \left(1 - \frac{d}{2}\right)G(\mathcal{T})\right]s^\mu\,,} \tag{6.11}$$

fixes the Neummann boundary conditions for the tilde-frame, $\delta\left(\sqrt{-\tilde{h}}\,\tilde{\mathcal{T}}^{ab}\right) = 0$, which may be written as following

$$\delta(\sqrt{-h}\,\mathcal{T}^{ab}) - \frac{G}{2}\delta(\sqrt{-h}h^{ab}) = -\frac{\sqrt{-h}}{2}\frac{\Omega'}{\Omega}\left[2\mathcal{T}^{ab} - h^{ab}\left(\mathcal{T} - \frac{d}{2}G + G\right)\right]\delta\mathcal{T}\,. \tag{6.12}$$

As expected, for $\Omega = 1$ (which implies $G = 0$), the above reduces to the Neumann boundary conditions, $\delta(\sqrt{-h}\,\mathcal{T}^{ab}) = 0$. In contrast, for a generic $\Omega(\mathcal{T})$, it becomes a mixture of Neumann and Dirichlet boundary conditions with $\mathcal{T}$-dependent coefficients, providing another example of freelance holography as a deformation of the Neumann boundary condition.

(7) The four classes in the previous section, Dirichlet (5.9), Neumann (5.14), conformal (5.17) and conformal conjugate boundary conditions are examples from our one-function family characterized by a number $k$ and represented by a $G$ linear in $\mathcal{T}$:

$$G(\mathcal{T}) = k\,\mathcal{T}\,, \qquad \Omega = B\,\mathcal{T}^b\,, \qquad b = \frac{2k}{2-dk}\,, \tag{6.13}$$

where the above is a result of (6.3) and $B$ is an inessential integration constant. Specific choices for $k = 0, 1$ and $k = \frac{2}{d}$ reproduce Dirichlet, Neumann, and conformal boundary conditions, respectively. For case $G(\mathcal{T}) = k\,\mathcal{T}$, we find the following relations,

$$\tilde{h}_{ab} = \mathcal{T}^{\frac{2k}{dk-2}} h_{ab}\,, \qquad \tilde{\mathcal{T}}^{ab} = \mathcal{T}^{\frac{(d+2)k}{2-dk}}\left[\mathcal{T}^{ab} - \frac{k}{2}\mathcal{T}\,h^{ab}\right]\,,$$
$$-\frac{1}{2}\int_\Sigma \mathrm{d}^d x\,\sqrt{-\tilde{h}}\,\tilde{\mathcal{T}}^{ab}\,\delta\tilde{h}_{ab} = \frac{1}{2}\int_\Sigma \mathrm{d}^d x\left[k\,h_{ab}\,\delta(\sqrt{-h}\,\mathcal{T}^{ab}) - (1-k)\,\sqrt{-h}\mathcal{T}^{ab}\,\delta h_{ab}\right]\,. \tag{6.14}$$

It shows that the Neumann boundary condition in the original frame ($k = 1$) is mapped onto the Dirichlet boundary condition in the tilde frame and vice-versa. For a generic $k$ it interpolates between Neumman and Dirichlet. Moreover, as noted in item (5) above, for the conformal boundary conditions $k = \frac{2}{d}$, the power $b$ blows up.

(8) As the last comment, we note that there exists a $W$ term

$$W^\mu_{\text{bulk}} = \frac{\sqrt{-g}}{d-1}\mathcal{T}\,s^\mu\,, \tag{6.15}$$

which yields the Lee-Wald (LW) symplectic potential [2]. The above $W$ term effectively removes the contribution of the Gibbons-Hawking-York boundary term in (5.1), leaving the Einstein-$\Lambda$ theory supplemented only by counterterms. The LW symplectic potential is

$$\boldsymbol{\Theta}_{\text{LW}} = \int_\Sigma \mathrm{d}^d x\left[-\frac{1}{2}\sqrt{-h}\,\mathrm{T}^{ab}\,\delta\mathrm{g}_{ab} + \frac{1}{d(d-1)}\mathcal{T}^{1-d}\delta(\sqrt{-h}\,\mathcal{T}^d) + \partial_a\left(\frac{\sqrt{-g}}{8\pi G}\,s^{[r}\delta s^{a]} + \frac{\Theta^a_{\text{ct}}}{8\pi G}\right)\right]$$
$$= \int_\Sigma \mathrm{d}^d x\left[\frac{1}{2}\mathrm{g}_{ab}\,\delta(\sqrt{-h}\,\mathrm{T}^{ab}) + \frac{1}{d(d-1)}\mathcal{T}^{1-d}\delta(\sqrt{-h}\,\mathcal{T}^d) + \partial_a\left(\frac{\sqrt{-g}}{8\pi G}\,s^{[r}\delta s^{a]} + \frac{\Theta^a_{\text{ct}}}{8\pi G}\right)\right]\,, \tag{6.16}$$

which yields the symplectic form[4]

$$\boldsymbol{\Omega}_{\text{LW}} = \int_\Sigma \mathrm{d}^d x\left[\frac{1}{2}\delta\mathrm{g}_{ab} \wedge \delta(\sqrt{-h}\,\mathrm{T}^{ab}) + \frac{1}{d}\delta(\sqrt{-h}) \wedge \delta\mathcal{T} + \partial_a\left(\frac{\delta(\sqrt{-g}\,s^{[r}) \wedge \delta s^{a]}}{8\pi G} + \frac{\delta\Theta^a_{\text{ct}}}{8\pi G}\right)\right]\,. \tag{6.17}$$

The distinction between the above and the symplectic form of all previous cases lies in the last term, which can be removed by the extra $Y$-term in those cases. Requiring the vanishing of the codimension-one part of $\boldsymbol{\Theta}_{\text{LW}}$ results in the following two classes of boundary conditions:

---

[4] We note that for metric (5.3) we have $\delta(\sqrt{-g}\,s^{[r}) \wedge \delta s^{a]} = -\frac{1}{2}\,\delta(\frac{\sqrt{-h}}{N}) \wedge \delta U^a$.

(i) $\delta g_{ab} = 0, \delta(\sqrt{-h}\,\mathcal{T}^d) = 0$. This is an extension of conformal boundary conditions in which $\delta(\sqrt{-h}\,\mathcal{T}) = 0$ is replaced with $\delta(\sqrt{-h}\,\mathcal{T}^d) = 0$. This boundary condition has been discussed in [89, 90]. We note that in the Weyl frame with $\Omega = \mathcal{T}^{-2}, G = \frac{2\mathcal{T}}{d-1}$, this boundary condition takes the form $\delta\tilde{g}_{ab} = \delta g_{ab} = 0, \delta(\sqrt{-\tilde{h}}) = 0$, which is the standard Dirichlet boundary conditions in the tilde-frame.

(ii) $\delta(\sqrt{-h}\,\mathrm{T}^{ab}) = 0, \delta(\sqrt{-h}\,\mathcal{T}^d) = 0$. This represents a new type that has not been explored in the literature. In the Weyl frame with $\Omega = \mathcal{T}^{-2}, G = \frac{2\mathcal{T}}{d-1}$, this boundary condition takes the form $\delta(\sqrt{-\tilde{h}}\,\tilde{\mathrm{T}}^{ab}) = 0, \delta(\sqrt{-\tilde{h}}) = 0$, which is the conjugate conformal boundary conditions discussed in section 5.4 in the tilde-frame.

## 6.2   Dual boundary description

According to the AdS/CFT duality, an AdS bulk theory with Dirichlet boundary conditions corresponds to a conformal field theory at the AdS boundary, which is a fixed point in the RG space of the boundary field theories. In section 3, we demonstrated that modifying bulk boundary conditions via $W$-freedom induces boundary deformations by generic multi-trace operators. The deformations associated with the boundary conditions studied in Section 5 are marginal single-trace deformations proportional to the trace of the boundary energy-momentum tensor (rBY-EMT). Those in this section, besides the single trace deformations (discussed in item (7) above), in general, include multi-trace deformations by a function of the trace of the rBY-EMT $\mathcal{T}$.

However, in non-anomalous CFTs, the EMT is traceless. Consequently, if we start with a CFT at the conformal fixed point and introduce a deformation that depends on the trace of the EMT, the boundary theory remains unchanged due to boundary conformality. To obtain non-trivial deformations, we must consider either (1) anomalous CFTs or (2) non-marginal deformations (deformations away from the conformal fixed point).

**Anomalous CFTs.**   CFTs on a generic manifold $\Sigma$ in even dimensions exhibit conformal anomalies. For instance, in $d = 2$, the trace of the energy-momentum tensor is proportional to the Ricci scalar of $\Sigma$ times the central charge of the CFT [91]. In $d = 4$, the trace anomaly is expressed in terms of the Euler density and the squared Weyl tensor of the background [91]. Consequently, in the presence of a non-flat boundary geometry, the energy-momentum tensor acquires a nonvanishing trace. In such cases $\mathcal{T}$ is not exactly marginal and all deformations considered in this section lead to nontrivial deformation of the boundary theory.

**Non-marginal deformations.**   In the absence of trace anomaly, deformations by the trace of the energy-momentum tensor at the conformal fixed point are trivial. However, one can move away from the fixed point by introducing other non-marginal deformations and then turn on deformations by the trace of the energy-momentum tensor. For instance, if we first deform the theory with a $T\bar{T}$-like deformation [92, 93], the resulting theory will no longer be a CFT, and the deformed energy-momentum tensor will acquire a nonvanishing trace and consequently the deformations by $\mathcal{T}$ become nontrivial. In such cases the gravity (bulk) theories with different boundary conditions become distinguishable. Notably, the magnitude of the initial deformation is irrelevant—once the trace is nonzero, even a small trace deformation can have significant effects.

We conclude this section with comments on the well-posedness of the boundary conditions introduced in this paper at a finite cutoff. It is worth emphasizing that, while our work primarily addresses asymptotic

boundaries, the boundary conditions, and corresponding deformations introduced here can also be applied to boundaries at a finite distance. This question will be discussed in detail in a companion paper and here we just make some comments. In [89], a one-parameter family of boundary conditions, labeled by $p$, was introduced. This family represents a special case of our more general boundary conditions, where the $W$ term is linear in $\mathcal{T}$.[5] We examined this specific family in detail in item (7) of the previous subsection. The parameter $p$ in their formulation is related to our parameter $b$ (6.13) by the following expression:

$$p = -\frac{1}{bd}. \tag{6.18}$$

The analysis in [89] is focused on four-dimensional Einstein gravity ($d = 3$) with a boundary at a finite distance. They found that static, spherically symmetric Euclidean solutions with boundary conditions corresponding to $p < 1/6$ ($b < -2$) exhibit both dynamical and thermodynamic instabilities. In contrast, for $p > 1/6$ ($b > -2$), these solutions are dynamically stable and the critical value $p = 1/6$ ($b = -2$) is marginally stable.

In generic $d$ one can show that $b = -2$ is the marginally stable case with $b > -2$ being stable and $b < -2$ being unstable. We note that (6.13) implies $b \geq -2$ corresponds to $k \leq \frac{2}{d}$ or $k \geq \frac{2}{d-1}$. More specifically, $k \leq \frac{2}{d}$ corresponds to $b > -\frac{2}{d}$ and $k \geq \frac{2}{d-1}$ to $-2 \leq b < -\frac{2}{d}$. Remarkably, the critical $b = -2$ ($k = \frac{2}{d-1}$) case is exactly the case discussed in item (8) at the end of the previous section. For this case, the gravity theory without any boundary term is mapped onto a Dirichlet (or conjugate conformal) boundary condition in the tilde frame.

# 7 Discussion and outlook

The AdS/CFT duality is traditionally formulated with Dirichlet boundary conditions at the AdS boundary on the gravity (bulk) side [8, 9, 75]. A natural question then arises: How can this duality be extended to accommodate arbitrary bulk boundary conditions? This question lies at the heart of the *freelance holography* program. In this work, we have explored this question while keeping the location of the boundary conditions imposed at the AdS causal boundary, allowing for arbitrary boundary conditions. In a companion work, we extend the freelance holography by permitting the boundary to be an arbitrary timelike codimension-one surface inside AdS, while also allowing arbitrary boundary conditions.

In generally covariant gravitational theories, modifying boundary conditions necessitates introducing appropriate boundary Lagrangians. The Covariant Phase Space Formalism (CPSF) provides a systematic framework for incorporating such boundary terms. Within this formalism, we have put forward the freelance holography program in the large-$N$ limit, thereby broadening the standard gauge/gravity perspective on boundary conditions in holography.

A key aspect of our construction and the usual AdS/CFT is the role of regularity conditions and bulk interior boundary conditions. In the standard AdS/CFT correspondence, asymptotic data alone are insufficient to fully determine duality. To construct a complete holographic dictionary, additional constraints from the bulk interior must be taken into account [12, 75]. This necessity becomes clear from the fact that an asymptotic expansion alone cannot distinguish between pure AdS space and an AdS black hole. Thus, a proper dual description requires additional information from the bulk interior. Although the regularity condition is a fundamental ingredient in our derivations, its role and implications within the CPSF, particularly in the presence of interior horizons, should be further explored in future research.

---

[5]As mentioned earlier, our definition of the conformal boundary condition slightly differs from that in [89], where it is imposed as $\delta K = \delta \hat{\tilde{\mathcal{T}}} = 0$; see the discussion below equation (5.16).

A key motivation for this study is to understand the dual interpretation of the varying bulk boundary conditions within the boundary theory. In this work, beginning with the standard gauge/gravity correspondence, we have provided a concrete derivation of Witten's proposition [27] that multi-trace deformations of the boundary theory correspond to changing boundary conditions on bulk fields within the CPSF framework. We have explicitly constructed the corresponding boundary Lagrangian that enforces the desired boundary conditions.

We have also demonstrated that $W$-freedom can modify the natural canonical conjugates of the phase space, effectively altering the parameterization of the solution space. This transformation, which is a canonical transformation in the CPSF and when surface charges and their algebra are concerned, is known as the change of slicing [21–26]. $W$ serves as the generator of canonical transformations on the solution phase space and governs the change of slicing.

Besides the $W$-freedom, CPSF also possesses another important freedom: the $Y$-freedom. We have discussed that the $Y$-freedom in the holographic context encodes all the information about the boundary symplectic potential, providing deeper insights into the role of boundary dynamics in holography. This is another central outcome of our study. A summary of the holographic interpretation of these freedoms in CPSF is presented in Table 1.

| Freedom | Bulk Interpretations | Boundary Interpretations |
|---|---|---|
| $W$-freedom | – Counterterms to ensure finite physical quantities<br>– Modification of bulk boundary conditions<br>– Generator of canonical transformations and change of slicing in the bulk solution phase space | – Regularization of boundary quantities<br>– Generator of multi-trace deformations |
| $Y$-freedom | Determining the codimension-two part of symplectic form | Determining the on-shell symplectic form |
| $Z$-freedom | Specifies corner Lagrangian of the bulk theory | Specifying boundary conditions and the slicing |

**Table 1**: Summary of the interpretations of $W$-, $Y$-, and $Z$-freedoms in bulk and boundary theories.

It is instructive to note that the boundary conditions introduced in this work differ slightly from the standard ones. In gravity, Neumann and conformal boundary conditions are traditionally defined by fixing the extrinsic curvature $K^{ab}$ or, equivalently, the unrenormalized BY-EMT $\mathring{\mathcal{T}}^{ab}$. However, in this work, we define Neumann and conformal boundary conditions by fixing the rBY-EMT $\mathcal{T}^{ab}$. This definition is physically well-motivated, as the rBY-EMT is regularized and finite, thus serving as the true canonical conjugate to the induced metric. As a consequence, by construction, the on-shell action and other physical quantities remain finite under these boundary conditions. This is a crucial advantage of our formulation since, for the standard Neumann and conformal boundary conditions the appropriate counterterms required to ensure a finite on-shell action and other observables are not well understood.

We close this paper with a list of future research directions.

- **Freelance Holography II [94]:** As discussed above, the freelance holography program should be pursued by relaxing the AdS causal boundary. This means imposing the relaxed boundary conditions on any arbitrary codimension-one timelike or null surface within the AdS interior. This project

aims to extend and generalize the $T\bar{T}$ deformation [92, 93], which holographically corresponds to the imposing of Dirichlet boundary conditions at a finite $r$ (associated with the UV cutoff of the boundary theory) [95, 96].

- **Well-posedness of boundary conditions:** In Section 6, we have constructed a family of hydrodynamic boundary conditions, specified by an arbitrary function of the trace of the rBY-EMT. A crucial and interesting question is the well-posedness of these boundary conditions [84]. It is well-known that boundary conditions at finite distances are more subtle than those at asymptotic boundaries. For example, Dirichlet boundary conditions at finite distances are generally not well-posed [83, 84, 89, 90, 97–100]. Therefore, the search for appropriate boundary conditions at finite distances is both necessary and important. As discussed, our family of one-function boundary conditions can be imposed at finite distances, making it a promising candidate for finding suitable well-posed boundary conditions. Specifically, the question arises: What properties must the function $\Omega$ satisfy to ensure a well-posed boundary condition? We provide brief comments on this at the end of Section 6.2, leaving a more detailed analysis for future work.

- **Classifying hydrodynamics deformations:** Our family of one-function boundary conditions has a hydrodynamic interpretation. In this framework, the canonical pairs describe a hydrodynamic system, and the tilted energy-momentum tensor is conserved with respect to the rescaled metric. An additional natural question that emerges, potentially guiding future research, is: What are the most general deformations for which the resulting canonical pairs retain a hydrodynamic interpretation?

- **Addition of matter fields.** Although our construction in Sections 2, 3, and 4 is general, in Sections 5 and 6 we specifically focus on boundary conditions and deformations in pure Einstein gravity in arbitrary spacetime dimensions. It would be valuable to extend this framework by incorporating matter fields (such as scalars, electromagnetism, etc.) and investigating how these fields influence the boundary conditions and matter deformations.

- **Other type of boundary conditions.** The CPSF-based construction in this work provides a framework for formulating boundary conditions beyond the four cases discussed in the paper. One such class of boundary conditions is the Robin (mixed) boundary conditions [101]. It is interesting to explore holography with the Robin and other possible "non-standard" boundary conditions.

## Acknowledgment

We would like to thank H. Adami, B. Banihashemi, E. Shaghoulian, and M.H. Vahidinia for their useful discussions.

## A    Example: free massive scalar field

To illustrate the CPSF framework and the associated freedoms discussed in Section 2.1, we consider a simple example: a free massive scalar field propagating in an AdS background [12]

$$S = \frac{1}{2} \int \mathrm{d}^{d+1}x \, \sqrt{|g|} \left( g^{\mu\nu} \partial_\mu \Phi \partial_\nu \Phi + m^2 \Phi^2 \right) + \int \mathrm{d}^{d+1}x \, \partial_\mu \mathcal{L}_{\mathrm{ct}}^\mu \,, \tag{A.1}$$

where $g_{\mu\nu}$ represents the Euclidean AdS metric

$$\mathrm{d}s^2 = g_{\mu\nu} \, \mathrm{d}x^\mu \, \mathrm{d}x^\nu = \ell^2 \left( \frac{\mathrm{d}\rho^2}{4\rho^2} + \frac{1}{\rho} \, \mathrm{d}x^i \, \mathrm{d}x^i \right) . \tag{A.2}$$

The above is related to (2.7) through $\rho = r^2/\ell$. The first variation of the action (A.1) yields

$$\delta S = \int \mathrm{d}^{d+1}x \left( \mathrm{E}\, \delta\Phi + \partial_\mu \Theta^\mu_{\text{bulk}} \right) , \tag{A.3}$$

The bulk equation of motion is obtained by setting $\mathrm{E} = 0$,

$$(-\Box_g + m^2)\Phi = 0 , \tag{A.4}$$

and the symplectic potential, $\Theta^\mu$, can be written as [6]

$$\Theta^\mu_{\text{bulk}} = \Theta^\mu_{\text{D}} + \delta W^\mu_{\text{bulk}} + \partial_\nu Y^{\mu\nu}_{\text{bulk}} , \qquad \Theta^\mu_{\text{D}} = \sqrt{-g}\, g^{\mu\nu}\, \partial_\nu \Phi\, \delta\Phi + \delta\mathcal{L}_{\text{ct}} , \tag{A.5}$$

where $W^\mu_{\text{bulk}}$ and $Y^{\mu\nu}_{\text{bulk}}$ represent the two freedoms in defining the symplectic potential (2.2) To better understand the solution space of the theory, we next solve the bulk equations of motion (A.4). We define $\Phi = \rho^{(d-\Delta)/2}\phi$, upon which (A.4) takes the form

$$[m^2 - \Delta(\Delta - d)]\phi - \rho\, \partial_i \partial^i \phi + 2(2\Delta - d - 2)\rho\, \partial_\rho \phi - 4\rho^2 \partial_\rho^2 \phi = 0 . \tag{A.6}$$

To simplify the analysis we showcase $\Delta = \frac{d}{2} + 1$. Plugging the series expansion

$$\phi(x, \rho) = \sum_{n=0}^{\infty} \left[ \phi_n(x) + \psi_n(x) \ln \rho \right] \rho^n , \tag{A.7}$$

into (A.6) we obtain $m^2 - \Delta(\Delta - d) = 0$, which has the solution $\Delta = \frac{1}{2}\left( d + \sqrt{d^2 + 4m^2} \right)$. For our specific case ($\Delta = \frac{d}{2} + 1$), this implies that the mass of the scalar field is $m^2 = -\left(\frac{d}{2}\right)^2 + 1$, which is above the BF bound [102] $m^2 \geq -\left(\frac{d}{2}\right)^2$. The coefficients in the expansion (A.7) can be obtained as follows:

$$\psi_0 = 0, \quad \psi_1 = -\frac{1}{4}\Box\phi_0, \quad \psi_2 = \frac{1}{32}\Box^2\phi_0, \quad \psi_3 = -\frac{1}{768}\Box^3\phi_0, \quad \psi_4 = \frac{1}{36864}\Box^4\phi_0, \quad \cdots$$

$$\phi_2 = -\frac{1}{8}\left( \Box\phi_1 + \frac{3}{8}\Box^2\phi_0 \right), \quad \phi_3 = \frac{1}{192}\left( \Box^2\phi_1 + \frac{7}{12}\Box^3\phi_0 \right), \quad \phi_4 = -\frac{1}{9216}\left( \Box^3\phi_1 + \frac{35}{48}\Box^4\phi_0 \right), \quad \cdots$$

where $\Box := \delta^{ij}\partial_i\partial_j$. The preceding equations show that all $\phi_n, \psi_n$ in (A.7) can be expressed in terms of $\phi_0$ and $\phi_1$. Consequently, the most general solution to the second-order differential equation (A.6) is parametrized by $\phi_0$ and $\phi_1$. Next, we explore regularity and imposing boundary conditions.

**Regularity in the interior.** The regularity condition in the interior establishes a relationship between $\phi_0$ and $\phi_1$, namely

$$\phi_1 = \phi_1[\phi_0] . \tag{A.8}$$

This relation is generally *non-local*. To see this, let us return to the wave equation (A.4) and perform a Fourier transform in the $x$-coordinates

$$\Phi(\rho, x) = \int \frac{\mathrm{d}^d k}{(2\pi)^d} e^{i\, k.x} \Phi(\rho, k) , \tag{A.9}$$

which transforms (A.4) into

$$-4\rho^2\, \partial_\rho^2 \Phi + 2(d-2)\rho\, \partial_\rho \Phi + k^2\, \rho\, \Phi + \Delta(\Delta - d)\Phi = 0 . \tag{A.10}$$

---

[6] In this section, we use the simplified notation $\Theta^\mu_{\text{D}}$ instead of $(\Theta^{\text{D}}_{\text{bulk}})^\mu$.

To simplify this equation, we introduce the change of variables $\rho = z^2$ and the field redefinition $\Phi = z^{d/2}\chi$, which transforms it into the modified Bessel equation

$$z^2 \partial_z^2 \chi + z \partial_z \chi - (k^2 z^2 + \nu^2)\chi = 0\,, \qquad \nu := \sqrt{m^2 + (d/2)^2}\,, \tag{A.11}$$

with two linearly independent solutions: the modified Bessel functions of the first kind, $I_\nu(kz)$, and the second kind, $K_\nu(kz)$. Since $I_\nu(kz)$ diverges as $z \to \infty$, regularity in the interior requires us to retain only the solution $\chi(z) \sim K_\nu(kz)$. For our special case, $\Delta = \frac{d}{2} + 1$, we have $\nu = 1$, and thus we consider $K_1(kz)$. Its asymptotic behavior as $z \to 0$ is

$$K_1(z) = \frac{1}{z} + \frac{z}{4}\left[(-1 + 2\gamma_{\mathrm{E}} - 2\ln 2) + 2\ln z\right] + \cdots\,, \tag{A.12}$$

where $\gamma_{\mathrm{E}}$ is Euler's constant with numerical value $\approx 0.577$. Now, let us express $\Phi$ in terms of the $\rho$ coordinate,

$$\Phi(\rho, k) = \rho^{(d-2)/4} \phi_0(k)\left[1 + \frac{k^2 \rho}{4}\left[(-1 + 2\gamma_{\mathrm{E}} + 2\ln(k/2)) + \ln \rho\right] + \cdots\right]. \tag{A.13}$$

The above expression contains only one arbitrary function, $\phi_0(k)$. The second arbitrary function, $\phi_1(k)$, in the asymptotic expansion (A.7) is fixed by the regularity condition in the bulk interior. More specifically, (A.13) implies

$$\phi_1(k) = \frac{k^2}{4}\left[(-1 + 2\gamma_{\mathrm{E}}) + 2\ln(k/2)\right]\phi_0(k)\,. \tag{A.14}$$

This explicitly demonstrates the non-local relationship between $\phi_0(x)$ and $\phi_1(x)$. More precisely,

$$\phi_1(x) = -\frac{\Box}{4}\left[(-1 + 2\gamma_{\mathrm{E}} - \ln 4) + \ln(-\Box)\right]\phi_0(x)\,. \tag{A.15}$$

This result shows that the relation between $\phi_0(x)$ and $\phi_1(x)$ involves an infinite number of derivatives and hence is non-local.

**On-shell action and regularization.** Consider the on-shell action

$$S \mathrel{\hat{=}} \frac{1}{2}\int \mathrm{d}^{d+1}x\, \partial_\mu \left(\sqrt{|g|}\, g^{\mu\nu}\Phi\, \partial_\nu \Phi + 2\mathcal{L}_{\mathrm{ct}}^\mu\right)\,, \tag{A.16}$$

where for the case $(\Delta = d/2 + 1)$, it takes the following form

$$S[\phi_0(x), \phi_1(x)] \mathrel{\hat{=}} \frac{1}{2}\int_\Sigma \mathrm{d}^d x \left[(d/2 - 1)\phi_0^2\, \epsilon^{-1} - \frac{d}{4}\phi_0 \Box \phi_0\, \ln \epsilon + \phi_0\left(d\,\phi_1 - \frac{1}{2}\Box \phi_0\right) + 2\mathcal{L}_{\mathrm{ct}}^\rho\right]\,, \tag{A.17}$$

as $\epsilon \to 0$. The first and second terms in the integral above are divergent. To regularize the on-shell action, these divergent terms must be absorbed into $\mathcal{L}_{\mathrm{ct}}^\mu$. Such boundary Lagrangians are referred to by different names depending on the context: in holographic renormalization, they are known as counterterms, whereas in the CPSF formalism, they are associated with a part of the $W$-freedom. The following boundary Lagrangian does the job:

$$\begin{aligned}
\int_\Sigma \mathrm{d}^d x\, \mathcal{L}_{\mathrm{ct}}^\rho &= -\frac{1}{2}\int_\Sigma \mathrm{d}^d x\, \sqrt{\gamma}\left[(d/2 - 1)\Phi^2 - \frac{1}{2}\ln \epsilon\, \Phi \Box_\gamma \Phi\right] \\
&= -\frac{1}{2}\int_\Sigma \mathrm{d}^d x\left[(d/2 - 1)\phi_0^2\, \epsilon^{-1} + (d-2)\phi_0\, \phi_1 - \frac{d}{4}\ln \epsilon\, \phi_0 \Box \phi_0\right]\,,
\end{aligned} \tag{A.18}$$

where $\gamma_{ij} := \frac{1}{\epsilon}\delta_{ij}$. Finally, we obtain a regularized on-shell action

$$S[\phi_0(x), \phi_1(x)] = \int_\Sigma \mathrm{d}^d x\, \phi_0\left(\phi_1 - \frac{1}{4}\Box \phi_0\right)\,. \tag{A.19}$$

For later use, we express the on-shell, regularized action in Fourier space as

$$S[\phi_0(k), \phi_1(k)] = \int \mathrm{d}^d k \, \phi_0^*(k) \left( \phi_1(k) + \frac{k^2}{4} \phi_0(k) \right) . \tag{A.20}$$

The regularity condition in the bulk relates $\phi_0(k)$ and $\phi_1(k)$ (A.14). Substituting this relation into the regularized on-shell action (A.20) yields

$$S[\phi_0(k)] = \int \mathrm{d}^d k \, \frac{k^2}{2} \left[ \gamma_{\mathrm{E}} + \ln(k/2) \right] \phi_0(k) \, \phi_0^*(k) . \tag{A.21}$$

**Symplectic potential.** We now compute the Dirichlet symplectic potential on the AdS boundary $\Sigma$

$$\Theta_{\mathrm{D}} = \int_\Sigma \mathrm{d}^d x \, \Theta_{\mathrm{D}}^\rho = \int_\Sigma \mathrm{d}^d x \left[ 2\rho^{1-d/2} \, \partial_\rho \Phi \, \delta\Phi + \delta\mathcal{L}_{\mathrm{ct}}^\rho \right] . \tag{A.22}$$

The total symplectic potential is given by (A.5) with $W_{\mathrm{bulk}}^\mu = 0$ and $Y_{\mathrm{bulk}}^{\mu\nu} = 0$, leading to $\Theta_{\mathrm{bulk}}^\mu = \Theta_{\mathrm{D}}^\mu$

$$\Theta_{\mathrm{bulk}}[\phi_0(x), \phi_1(x); \delta\phi_0(x), \delta\phi_1(x)] := \int_\Sigma \mathrm{d}^d x \, \Theta_{\mathrm{bulk}}^\rho = 2 \int_\Sigma \mathrm{d}^d x \left( \phi_1 - \frac{1}{4}\Box\phi_0 \right) \delta\phi_0 . \tag{A.23}$$

When transformed into Fourier space, it becomes

$$\Theta_{\mathrm{bulk}}[\phi_0(k), \phi_1(k); \delta\phi_0(k), \delta\phi_1(k)] = 2 \int_\Sigma \mathrm{d}^d k \left( \phi_1(k) + \frac{k^2}{4} \phi_0(k) \right) \delta\phi_0^*(k) . \tag{A.24}$$

Then by using (A.14) it results in

$$\begin{aligned}
\Theta_{\mathrm{bulk}}[\phi_0(k); \delta\phi_0(k)] &= \int_\Sigma \mathrm{d}^d k \, \left[ \gamma_{\mathrm{E}} + \ln(k/2) \right] k^2 \, \phi_0(k) \, \delta\phi_0^*(k) \\
&= \delta \int_\Sigma \mathrm{d}^d k \, \frac{k^2}{2} \left[ \gamma_{\mathrm{E}} + \ln(k/2) \right] \, \phi_0(k) \, \phi_0^*(k) \\
&= \delta S .
\end{aligned} \tag{A.25}$$

In the last line, we have applied (A.21). Interestingly, the on-shell symplectic potential for a scalar field, when subject to the regularity condition, simplifies to a total variation, which corresponds to the on-shell regularized action given in (A.21).

**Correlation functions.** Now we can simply read the $n$-point functions

$$\langle \mathcal{O}(k) \rangle = \frac{k^2}{2} \left[ \gamma_{\mathrm{E}} + \ln(k/2) \right] \phi_0^*(k) , \tag{A.26}$$

$$\langle \mathcal{O}(k) \, \mathcal{O}^*(k) \rangle = \frac{k^2}{2} \left[ \gamma_{\mathrm{E}} + \ln(k/2) \right] . \tag{A.27}$$

All higher $n$-point functions are vanishing.

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
