# Peer review of "Freelance Holography, Part I: Setting Boundary Conditions Free in Gauge/Gravity Correspondence"

_SciPost Physics_

## Round 2 · Referee Report · Anonymous (Referee 1) · 2025-6-8

Report

This article discusses the ambiguities in the covariant phase space formalism from the point of view of the gauge/gravity duality. In particular, one of the ambiguities corresponds to the choice of boundary conditions in the bulk. One motivation to study such ambiguities is to generalize the gauge/gravity duality usually formulated with Dirichlet boundary conditions for the bulk fields to more general types of boundary conditions. The main points of the paper are:

-there are three types of ambiguities/freedoms (W,Y,Z) in the covariant phase space formalism, which can be seen from the definition of the presymplectic potential. W corresponds to changing boundary conditions in the bulk/adding multi-trace deformations in the boundary theory. Y fixes the symplectic form for the boundary modes in the bulk/on-shell symplectic form in the boundary theory. Z corresponds to adding corner lagrangian terms in the bulk/changing boundary conditions in the boundary theory.

-the total symplectic potential (for concreteness in pure gravity) depends on the ambiguities and various choices lead to various types of boundary conditions such as Dirichlet, Neumann, conformal etc. The definition that the authors use for conformal is different from the usual one from literature because they first add the counterterms that make the on-shell action with Dirichlet boundary conditions finite and then change boundary conditions from Dirichlet to conformal.

-there exists a one-function family of boundary conditions (hydrodynamical Weyl frames), which contains and generalizes Dirichlet and conformal.

While many of the results that appear in the paper are not new, they are reformulated in the covariant phase space formalism language, in a unified way. Considering new types boundary conditions in holography is a promising direction of extending AdS/CFT.

I recommend this article to be published in SciPost. Some comments/questions for the authors:

-since one of the nice things about the covariant phase space formalism is that it provides a way to define conserved charges and their algebras, it would be good if the authors would comment on how these quantities depend on the ambiguities discussed

-in previous works like Setting the boundary free in AdS/CFT, authors would find counterterms that lead to finite quantities in the bulk for non-Dirichlet boundary conditions. In this paper, counterterms are considered for Dirichlet boundary conditions and then the boundary conditions are changed. It would be good if the authors could comment on the relation between the two.

-the analysis in this paper is in the saddle point approximation at large N, but the authors conjecture that can be extended beyond this approximation. Do the authors have in mind any ways to test their proposal?

-regarding the discussion in section 6.2 about non-marginal deformations: the analysis in the paper is focused on AdS/CFT. In the case when we first deform the boundary theory such that it is no longer a CFT, do the authors suggest that the analysis still holds? It would be interesting to add a comment about validity outside AdS/CFT.

Recommendation

Publish (meets expectations and criteria for this Journal)

---

## Round 2 · Referee Report · Anonymous (Referee 2) · 2025-6-30

Strengths

(1) Provides new insights into the AdS/CFT dictionary assuming more general boundary conditions on bulk fields.

(2) Clearly written

Weaknesses

(1) Few new results; many parts are review of existing literature.

Report

Developing a holographic description of gravitating regions of spacetime beyond asymptotically Anti-de Sitter (AdS) space remains an open problem. Crucial to this active line of inquiry is a careful treatment of the location of timelike boundaries and the boundary conditions imposed on said boundaries. The purpose of this article, the first in a two-part series, is to revisit aspects of AdS/CFT in the large-$N$ limit from the lens of the covariant phase space formulation of classical general relativity. In particular, this article interprets known ambiguities in the covariant phase formalism (e.g., the Jacobson-Kang-Myers ambiguities), dubbed ``freedoms'', in terms of freedom in the choice of boundary conditions imposed on bulk field variables. Via their analysis, the authors argue for the notion of ``freelance holography'', in which the bulk field boundary conditions are relaxed from the standard Dirichlet boundary conditions.

The submitted manuscript is clearly and pedagogically written, and provides some new insights into the AdS/CFT dictionary with more general boundary conditions. My only hesitation in recommending this article for publication is that much of it is review of existing literature, with few new results. Indeed, the authors admit in the introduction that their work ``extends the analysis in [33] (Compere, Marolf, ``Setting the boundary free in AdS/CFT'', 0805.1902) to more general boundary conditions and to beyond three dimensional bulk theories''. Nonetheless, the article provides useful and correct analyses of alternative boundary conditions, e.g., conformal boundary conditions, in the context of AdS/CFT in the large-$N$ limit, a timely and beneficial addition to the literature, especially in combination with the second article in their two-part sequence.

Therefore, I am pleased to recommend this article be accepted for publication -- once my minor comments/questions are addressed.

(1) The authors should highlight throughout the manuscript the specific new results, beyond stating that their work extends the analysis of [33].

(2) A relevant article analyzing (finite) conformal boundaries in asymptotically AdS$_{4}$ the authors should consider (and cite along with Refs. [87] and [88]) is [hep-th/2412.16305].

(3) In Section 5.3, the authors consider boundary conditions that differ from standard conformal boundary conditions (CBCs). Standard CBCs are those are those which fix both the conformal class of the induced boundary metric and the trace of the extrinsic curvature. Instead, the authors characterize their CBCs are fixing the conformal class of the induced boundary metric and the trace of the renormalized Brown-York stress-tensor, their reasoning being that this will ensure finiteness of the on-shell action. I have two points of confusion here that I would like clarified.

(i) First, it seems the authors rely on working with a Gibbons-Hawking-York boundary term suitable for making the variational problem well-posed for Dirichlet boundaries (see Eq. (5.1)). When CBCs are imposed, however, it is known that the necessary Gibbons-Hawking-York boundary term differs by an overall constant factor. The authors should explain why they use the Gibbons-Hawking-York term appropriate for Dirichlet boundaries here, even though they impose different CBCs.

(ii) Related, imposing the standard CBCs leads to a different (unrenormalized) Brown-York stress-tensor (cf. Eqs. (103) and (106) of [hep-th/2409.07643]). From the perspective of CBCs, would it not be more natural to consider the Brown-York stress-tensors associated with CBCs?

Upon addressing (1) -- (3), I would be happy to recommend the article for publication in Scipost.

Requested changes

(1) Highlight the new results of the manuscript.

(2) Update references, e.g., cite [hep-th/2412.16305] along with Refs. [87] and [88]).

(3) Respond to points (i) and (ii) of question (3) of the report.

Recommendation

Ask for minor revision

---

## Editorial Decision

resubmitted